# ^α^*N*-Acetyl β-Endorphin Is an Endogenous Ligand of σ1Rs That Regulates Mu-Opioid Receptor Signaling by Exchanging G Proteins for σ2Rs in σ1R Oligomers

**DOI:** 10.3390/ijms24010582

**Published:** 2022-12-29

**Authors:** Javier Garzón-Niño, Elsa Cortés-Montero, María Rodríguez-Muñoz, Pilar Sánchez-Blázquez

**Affiliations:** Neuropharmacology, Department of Translational Neuroscience, Cajal Institute, CSIC, 28002 Madrid, Spain

**Keywords:** ^α^*N*-acetyl beta endorphin, sigma receptor types 1 and 2, mu-opioid receptor, beta endorphin 28–31, G-protein signaling, epsilon receptor

## Abstract

The opioid peptide β-endorphin coexists in the pituitary and brain in its ^α^*N*-acetylated form, which does not bind to opioid receptors. We now report that these neuropeptides exhibited opposite effects in in vivo paradigms, in which ligands of the sigma type 1 receptor (σ1R) displayed positive effects. Thus, ^α^*N*-acetyl β-Endorphin reduced vascular infarct caused by permanent unilateral middle cerebral artery occlusion and diminished the incidence of *N*-methyl-D-aspartate acid-promoted convulsive syndrome and mechanical allodynia caused by unilateral chronic constriction of the sciatic nerve. Moreover, ^α^*N*-acetyl β-Endorphin reduced the analgesia of morphine, β-Endorphin and clonidine but enhanced that of DAMGO. All these effects were counteracted by β-Endorphin and absent in σ1R^−/−^ mice. We observed that σ1Rs negatively regulate mu-opioid receptor (MOR)-mediated morphine analgesia by binding and sequestering G proteins. In this scenario, β-Endorphin promoted the exchange of σ2Rs by G proteins at σ1R oligomers and increased the regulation of G proteins by MORs. The opposite was observed for the ^α^*N*-acetyl derivative, as σ1R oligomerization decreased and σ2R binding was favored, which displaced G proteins; thus, MOR-regulated transduction was reduced. Our findings suggest that the pharmacological β-Endorphin-specific epsilon receptor is a σ1R-regulated MOR and that β-Endorphin and ^α^*N*-acetyl β-Endorphin are endogenous ligands of σ1R.

## 1. Introduction

Endorphins, a series of endogenous opioids, were named after the pituitary peptide β-endorphin 1–31 (β-End 1–31), which was discovered in 1976 [1]. This untriakontapeptide contains the methionine enkephalin sequence in its N-terminal sequence and produces analgesia through opioid receptors in the midbrain and spinal cord, which mainly include mu-opioid receptors (MORs) and, to a lesser extent, delta-opioid receptors (DORs). Despite its relevance as a pituitary and neuromodulatory hormone, the physiological significance of its ^α^*N*-acetylated form, which is also liberated from the intermediate lobe [2], as well as the possible existence of a β-End 1–31 specific receptor (the epsilon (ε) receptor), have remained elusive.

β-End 1–31 is derived from the carboxyl-terminal 31 amino acids of β-Lipotropin, and because it exhibits opioid activity, β-Lipotropin was named proopiomelanocortin (POMC) and is considered among the three families of endogenous opioid peptide precursors [3]. β-End 1–31 coexists with two shorter fragments, β-End 1–27 and β-End 1–26 [4], and with the ^α^*N*-acetylated forms of β-End 1–31, 1–27 and 1–26 [5]. The processing of β-End 1–31 into 1–27 generates the C-terminal peptide 28–31 (KKGE), from which the 30–31 dipeptide originates [6]. Most likely, the C-terminal peptide 27–31 (YKKGE) is derived from β-End 1–26, also producing 28–31 after Y27 is removed.

β-End 1–31 and its derivatives are present in the pituitary and discrete areas of the brain [6,7,8,9]. In in vitro assays, β-End 1–27 exhibits a lower affinity for the MOR than that of the parent 1–31 form [4] and produces weak analgesic effects in rodents [4,10,11]; thus, β-End 1–27 antagonizes MOR-mediated β-End 1–31 analgesia [12]. β-End 1–26 also displays lower affinity for MOR than that of β-End 1–31, and it is weaker than β-End 1–27, antagonizing the analgesia evoked by β-End 1–31. On the other hand, the β-End C-terminal dipeptide Gly-Gln (30–31) does not bind to opioid receptors but regulates certain nonopioid effects of β-End 1–31 [13]. The ^α^*N*-acetylated forms of β-End 1–27 and β-End 1–26 are also inactive in opioid binding assays and do not exhibit analgesic properties [5]; however, they regulate the antinociceptive activity of different MOR-binding opioids [14]. The ^α^*N*-acetyl β-End 1–31 peptide also regulates antinociception mediated by α2 adrenoceptors but not by DORs [15,16]. Interestingly, ^α^*N*-acetyl β-End 1–31 alleviates morphine withdrawal syndrome in rodents [17], and this effect is also observed for the β-End 1–31 C-terminal dipeptide 30–31 [18]. 

When inhibiting the electrically induced contraction of the isolated rat vas deferens (RVD), MOR-binding opioids exhibited a profile suggestive of the ε receptor, a specific receptor for β-End 1–31 [19,20]. With respect to the central nervous system, β-End 1–27 antagonizes β-End 1–31 supraspinal analgesia, but it does not interfere with the analgesia of agonists that act at MOR, DOR or kappa-opioid receptors (KOR), such as DAMGO, morphine, DPDPE, DADLE and U-50,488H [21]. These observations and other studies have suggested the existence of the specific ε receptor for β-End 1–31 in different brain areas [22,23,24]. In the presence of antagonists of the three opioid receptors MOR, DOR and KOR, β-End 1–31 still promotes a fraction of the initial GTPγS binding to G proteins, which is only abolished by its putative antagonist β-End 1–27. However, in neural membranes from triple MOR, DOR, and KOR knockout mice β-End 1–31 does not stimulates GTPγS incorporation to Gα subunits [25]. In this scenario, endogenous and exogenous agents modify the signaling activity of MORs by binding to accessory sites [26]. These observations suggested that the pharmacological ε receptor comprises a G-protein-coupled receptor [27], probably the MOR, and an accessory site that regulates the activity of the opioid receptor. The sigma type 1 receptor (σ1R) may be a candidate for such regulation. σ1R can be found in the neural membrane [28,29], and was initially considered an opioid receptor for benzomorphans, such as SKF-10,047 [30]. Later, σ1R was identified as part of a tonic antiopioid system [31] that is mostly linked to MOR analgesic effects. Compounds that bind to σ1Rs and enhance MOR analgesic effects are named antagonists, and agonists prevent such effects [32]. As expected, in σ1R^−/−^ mice, morphine antinociception cannot be regulated by σ1R ligands [28].

While β-End 1–31 exhibits agonist activity in the RVD, benzomorphans antagonize its effects [33], and in binding assays, the ε receptor was defined as a benzomorphan binding site that was distinct from canonical MOR, DOR or KOR [34,35,36]. Remarkably, benzomorphans, such as pentazocine and SKF-10,047, promote the separation of the σ1R from the 78-kDa glucose-regulated GRP78/BiP [37]; thus, these benzomorphans behave as agonists at σ1Rs, which exert a negative influence on MOR-mediated analgesia [31]. 

Our aim was to study the activity of β-End 1–31, its physiological fragments and ^α^*N*-acetylated derivatives in a series of in vivo, ex vivo and in vitro paradigms in which classical ligands of σ1R display effects. In these assays, β-End 1–31 exhibited σ1R agonist activity while ^α^*N*-acetyl β-End 1–31 behaved as an σ1R antagonist. In in vitro assays, β-End 1–31 and ^α^*N*-acetyl β-End 1–31 exhibited opposite effects on the association of σ1Rs with σ2Rs, NR1 C1 subunits of *N*-methyl-D-aspartate receptors (NMDARs) and BiP. Moreover, β-End 1–31 and its ^α^*N*-acetyl derivative regulated G protein binding to σ1Rs, and modified the signaling pathways regulated by MORs. It is possible that the pharmacologically named ε receptor is just a fraction of the MORs associated with σ1Rs so that β-End 1–31 binds to them with a high affinity and selectivity.

## 2. Results

### 2.1. ^α^N-Acetyl β-End 1–31 Diminishes the Neural Damage Caused by Permanent Unilateral Middle Cerebral Artery Occlusion (pMCAO)

In this animal model, surgery or the intracerebroventricular (icv) procedure does not significantly change the total brain volume [38]. However, 48 h after pMCAO, the mice showed severe tissue injury. This damage was most apparent in the cerebral cortex, and the infarct volume was estimated to affect 5.4 ± 1.2% of the total brain volume. No damage was observed in the sham-operated mice. Selective σ1R antagonists, such as S1RA and BD1063, exhibit protective effects in this animal model for stroke, which are prevented by σ1R agonists [38,39]. The icv administration of β-End 1–31 or the σ1R agonist PRE084 60 min postsurgery did not alter the extent of infarct volume; however, ^α^*N*-acetyl β-End 1–31 (3 nmol, icv) greatly reduced stroke outcomes, and much less severe infarction was observed (an approximate 80% reduction in the infarct size to 1.2 ± 0.9% of the total brain volume) after permanent cerebral ischemia. Notably, β-End 1–31 and PRE084 prevented the neuroprotection provided by ^α^*N*-acetyl β-End 1–31, thus suggesting the implication of σ1Rs in its effects. Accordingly, ^α^*N*-acetyl β-End 1–31 did not exhibit protective effects on pMCAO-produced ischemia in σ1R^−/−^ mice (Figure 1A).

### 2.2. Anticonvulsant Activity of ^α^N-Acetyl β-End 1–31

The anticonvulsant activity of β-End 1–31 and its ^α^*N*-acetyl derivative was studied in mice, in which seizures were promoted by icv injection of 1 nmol NMDA [40,41]. NMDA induced a series of anomalous behaviors in all the mice, such as tonic convulsions, compulsive rearing, wild running, clonic convulsions, tonic seizures, and, in approximately 15–20% of the animals, death [38,42] (Figure 1B). In this paradigm, antagonists at σ1Rs alleviated the incidence of the convulsive syndrome that follows NMDA administration [42]. The σ1R agonist PPCC or β-End 1–31 did not significantly alter the behavioral effects evoked by NMDA administration. In contrast, ^α^*N*-acetyl β-End 1–31 partially protected the mice from hypermobility and convulsive rearing, while only a tendency to diminish clonic convulsions and tonic seizures was observed, suggesting that distinct neural substrates underlie distinct components of the seizure model. PPCC prevented ^α^*N*-acetyl β-End 1–31 from alleviating these signs of NMDA-induced convulsive syndrome. These positive effects caused by ^α^*N*-acetyl β-End 1–31 were absent when the syndrome was modeled in σ1R^−/−^ mice, indicating that σ1R is necessary for the Y1-acetylated derivative of β-End 1–31 to produce the beneficial effects described above.

### 2.3. Anti-Allodynia Effect of ^α^N-Acetyl β-End 1–31

Unilateral chronic sciatic nerve constriction injury (CCI) promotes neuropathic pain in CD1 mice. Maximal mechanical allodynia was observed at approximately seven days postsurgery (as measured with the von Frey test) [43,44]. Allodynia is mostly detected in the ipsilateral operated paw and minimally impacts the response of the contralateral paw. Later, the animals slowly recovered their presurgery responses (Figure 2A). A low-icv dose of 0.3 nmol ^α^*N*-acetyl β-End 1–31 significantly reduced mechanical allodynia in these mice, and 3 nmol practically restored the von Frey response to the levels observed presurgery. The beneficial effect of ^α^*N*-acetyl β-End 1–31 was observed 10 to 15 min after icv administration, in which the higher dose clearly provided protection for 24 h and provided partial protection for an additional 24 h. The icv administration of 3 nmol PRE084, a σ1R agonist, together with 3 nmol ^α^*N*-acetyl β-End 1–31, prevented the β-End 1–31 ^α^*N*-acetyl derivative from diminishing the response of CCI mice in the von Frey test.

### 2.4. ^α^N-Acetyl Derivatives and C-Terminal Sequences of β-End 1–31 Diminished Supraespinal Analgesia Produced by Morphine and β-End 1–31 in Mice

The opioids morphine, DAMGO and β-End 1–31, which mostly act at MORs, and clonidine by acting at α2 adrenoceptors, injected by icv route promote supraspinal analgesia in mice, as determined by the tail-flick test with warm water nociceptive stimulus. ^α^*N*-acetyl β-End 1–31 injected icv in mice reduced the analgesic activity of morphine and β-End 1–31, while that of DAMGO and clonidine increased. The regulatory effects of ^α^*N*-acetyl β-End 1–31 on the analgesia of the aforementioned substances reached a maximum at remarkably low icv doses (1 to 10 fmols), and persisted when the dose was increased to 1 pmol or 1 nmol (Appendix A); furthermore, the effect of a single dose of ^α^*N*-acetyl β-End 1–31 lasted for 24 h, suggesting a hormonal regulatory role for this pituitary peptide [14,15]. Since ^α^*N*-acetyl β-End 1–31 does not bind or binds poorly to MOR [14,45,46], its regulatory effects seem to be produced through a saturable allosteric site that displays a very high affinity.

With this background, we explored whether ^α^*N*-acetyl β-End 1–31 and its derivatives interact with σ1Rs to regulate the analgesic activity of MOR and likely other GPCRs. Thus, icv administration of 1 pmol ^α^*N*-acetyl β-End 1–31 30 min before morphine and β-End 1–31 brought about significant reductions in their analgesic potency. In this paradigm, the σ1R agonist PPCC prevented the impairing effects of ^α^*N*-acetyl β-End 1–31 on morphine and β-End 1–31 analgesia. Mice with targeted disruption of the *σ1R* gene showed an increased analgesic response to both opioids, which was not altered by ^α^*N*-acetyl β-End 1–31 (Figure 2B).

β-End 1–27 and β-End 1–26, the derivatives of β-End 1–31 without 4 and 5 C-terminal residues, exhibit binding to MOR [4,12], and β-End 1–27 particularly diminishes the analgesia of the parent peptide but not that of morphine [21]. We observed that their ^α^*N*-acetyl derivatives, which do not bind to opioid receptors or promote analgesia [5], retained some activity of ^α^*N*-acetyl β-End 1–31 and antagonized β-End 1–31 analgesia (Figure 3A). However, the nonphysiological β-End 1–16 and its ^α^*N*-acetyl derivative practically did not alter β-End 1–31 analgesia. (Appendix A). Therefore, as the C-terminal sequence of β-End 1–31 is reduced from residues 27 and 26 to 16, their capacity to reduce β-End 1–31 analgesia diminished.

The processing of β-End 1–31 into β-End 1–26 and β-End 1–27 provides β-End 27–31 (YKKGE) and β-End 28–31 (KKGE) and the resulting dipeptide β-End 30–31 (GE). Notably, these C-terminal products reduced the analgesic activity of β-End 1–31 and morphine. Their antagonist activity was prevented by PPCC and did not occur in σ1R^−/−^ mice (Figure 3B). The nonphysiological C-terminal sequence β-End 22–31 reduced β-End 1–31 analgesia, and this effect was not produced by β-End 6–31, which lacks the N-terminal met-enkephalin 1–5 sequence (Appendix A).

The analgesia of β-End 1–31 was diminished by σ1R antagonists such as S1RA and BD1047; however, σ1R agonists PRE084 and PPCC did not alter the effect of β-End 1–31 (Figure 3C). Notably, ^α^*N*-acetyl β-End 1–31 impaired the analgesia of β-End 1–31 and morphine (Figure 2B); however, S1RA reduced β-End 1–31 analgesia and enhanced morphine analgesia (Figure 3D). The regulation by classical σ1R antagonists and ^α^*N*-acetyl β-End 1–31 of MOR analgesia was absent in σ1R^−/−^ mice. These observations suggested the implication of σ1Rs in the analgesic activity of β-End 1–31, and showed that ^α^*N*-acetyl β-End 1–31 and S1RA exhibit differences in their regulation of the MOR; both compounds reduced β-End 1–31 analgesia, ^α^*N*-acetyl β-End 1–31 reduced but S1RA enhanced morphine analgesia, and ^α^*N*-acetyl β-End 1–31 and S1RA enhanced DAMGO analgesia. On the other hand, neither ^α^*N*-acetyl β-End 1–31 nor S1RA altered the analgesic effects of the DOR agonists DPDPE and D-Ala^2^ Deltorphin II (Figure 3E). The effect of the ^α^*N*-acetylated peptide on morphine and β-End 1–31 was constant over various logs of doses (Appendix A). In contrast, S1RA produced a biphasic effect on morphine analgesia, 3 nmol produced the maximum increase and 10 nmol barely altered morphine effects (Figure 3D). 

### 2.5. σ1R Binds to Gi/o/q-11/z Proteins: Effect of β-End 1–31 and ^α^N-Acetyl β-End 1–31

In the absence of σ1Rs, the antinociception mediated by MORs significantly increases [28]. Our study confirmed that the supraspinal analgesia of icv DAMGO, a MOR agonist, was augmented in σ1R^−/−^ mice; however, the effects of icv DPDPE, a DOR agonist, or those of WIN55,212–2, a cannabinoid agonist at CB1 receptors, did not significantly change (Figure 4A). This finding correlated with an increased association of MOR with G proteins in σ1R^−/−^ mice (Figure 4B). The σ1R belongs to a tonic anti-opioid endogenous system [31], and thus, we investigated whether σ1Rs may reduce the availability of G proteins that are regulated by MORs. 

In ex vivo fishing assays performed on mouse solubilized brain membranes enriched in synaptosomes (SBM), our bait, agarose-covalently attached σ1Rs (agarose-σ1R), captured diverse prey, such as trimeric G proteins, σ1Rs and σ2Rs (Figure 4C). Because σ1Rs form oligomers in vivo that can be altered by ligands [47,48], we added in vitro free σ1Rs to agarose-σ1Rs. After removing the unbound σ1Rs, the extent of the σ1R oligomerization that remained around the agarose-σ1R was determined by SDS-PAGE followed by Western blot analysis (Appendix A). The ^α^*N*-acetyl β-End 1–31 and β-End 28–31 released σ1Rs from these oligomers, but β-End 1–31 strengthened these self-associations (Appendix A).

The G proteins included in the study are representative of those regulated by MORs in the production of opioid analgesia when injected by the icv route [49,50,51]. Thus, from the Gαi family [52] Gαi2 and Gαo, pertussis toxin (PTX)-sensitive, and Gαz, PTX-insensitive, were selected [53]. From the Gαq subfamily, the PTX-insensitive Gαq was selected. In ex vivo fishing assays performed with the generated σ1R oligomers, β-End 1–31 barely released Gi2 proteins but promoted the association of Gz and, to a greater extent, that of Go proteins. Notably, ^α^*N*-acetyl β-End 1–31, which displays an affinity toward σ1Rs similar to that of β-End 1–31, efficiently released G proteins from agarose-σ1Rs. The Go, Gq-11 (antibodies do not distinguish between Gqα and G11α subunits) and Gi2 proteins were released from σ1R oligomers with fM concentrations of ^α^*N*-acetyl derivative, while pM concentrations were needed for Gz to be freed. The σ1R antagonist S1RA, which displays a much weaker affinity for σ1Rs than that of ^α^*N*-acetyl β-End 1–31, also exhibited G-protein-releasing effects. The σ1R agonist PRE084 at the concentrations studied did not disrupt G protein binding to σ1R oligomers but increased the binding to Gz proteins (Figure 4D). On the other hand, β-End 1–26, 1–27 and the C-terminal tetrapeptide β-End 28–31 showed G-protein-releasing effects, while the dipeptide β-End 30–31 was much weaker (Appendix A).

Therefore, β-End 1–31 promotes the capture of G proteins, increases or stabilizes σ1R oligomers, and diminishes the presence of σ2Rs at the agarose-σ1R bait. In contrast, ^α^*N*-acetyl β-End 1–31 releases G proteins, reduces the extent of σ1R oligomerization, and increases the association of σ2Rs with the bait (Figure 4D and Figure 5A). Endogenous σ2R is not necessary for agarose-σ1Rs to bind to endogenous G proteins, although in SBM from σ2R^−/−^ mice, the capture diminished with respect to that obtained on wild-type SBM (Figure 5B). Remarkably, in the absence of σ2Rs, ^α^*N*-acetyl β-End 1–31 failed to release G proteins from σ1R oligomers (Figure 5C), and MORs coprecipitated fewer G proteins than that in wild-type mice (Figure 5D). In fact, σ2R^−/−^ mice exhibit an impaired analgesic response to MOR agonists [54]. 

In a set of assays, β-End 1–31, ^α^*N*-acetyl β-End 1–31, and S1RA were administered icv into mice, and the coprecipitation of G proteins, σ2Rs and NMDAR NR1 C1 subunits with MOR was studied. The subanalgesic dose of 5 pmol β-End 1–31 increased the association of MOR with the above-mentioned σ1R interactors, mostly with σ2Rs, Go, and Gi2 proteins and to a lesser extent with Gz proteins and NMDAR NR1 C1 subunits. This profile was slightly reproduced by ^α^*N*-acetyl β-End 1–31 and S1RA 3 nmol, while 10 nmol S1RA increased MOR association with Gi2/o and σ2Rs (Figure 5E).

The aforementioned observations suggest that both types of sigma receptors may physically interact in the MOR and NMDAR environments. This issue was addressed in vivo using Chinese hamster ovary (CHO) cells. The cells were transiently transfected with pSF-cMyc-σ1R and pSF-HA-σ2R constructs, grown on coverslips, fixed and immunolabeled with antibodies conjugated to the fluorescent dyes cMyc (Alexa Fluor 488) and HA (Alexa Fluor 647). Forty-eight hours after transfection, the chimeric proteins were detected in approximately 50% of the cells (Figure 6A,B). The sigma receptors appeared as a diffuse cytoplasmic stain accompanied by bright punctate spots of fluorescence that were suggestive of some receptor clustering at the plasma membrane (Figure 6C–F). In most cells, their punctate staining colocalized, which was compatible with the σ1R-σ2R physical interaction.

To ascertain that the sigma receptors were successfully expressed, we performed Western blot analysis of cell lysates using anti-tag-directed antibodies. The protein constructs were detected at 26/27 kDa for cMyc-σ1R and 20/21 kDa for HA-σ2R (Figure 6G). Most interestingly, immunoprecipitated σ1Rs were accompanied by σ2Rs (Figure 6H, Lane 1). We then explored the possibility of modifying this association with σ1R ligands. The substances were added to the cultures before the cell lysates were obtained. We observed that β-End 1–31 and the σ1R agonist PRE084 strongly decreased the σ1R-σ2R association, while ^α^*N*-acetyl β-End 1–31 and the σ1R antagonists S1RA and BD1047 preserved these protein complexes (Figure 6H). Peptides, such as β-End 1–31, do not penetrate freely into cells; therefore, σ1R ligands probably interact with σ1R-σ2R complexes at the plasma membrane or close surface structures. Indeed, σ1R ligands bind to brain synaptosomes with nM affinity [29,55], which is compatible with the σ1R colocalization with MORs and NMDARs in plasma membrane structures of periaqueductal brain neurons [28]. 

### 2.6. In Vitro Activity of β-End 1–31 and Physiological Derivatives on σ1R Associations with σ2Rs, NMDAR NR1 C1 Subunits and BiP

σ1R establishes regulatory interactions with a series of signaling proteins. Thus, in the MOR and glutamate NMDAR environment, σ1R associates with the cytosolic C1 region of NMDAR NR1 subunits [28,56] and, as observed here, with σ2Rs. In the endoplasmic reticulum (ER), σ1R associates with BiP [37]. The in vitro interactions of σ1Rs with these signaling proteins were addressed following the procedure described (Appendix A).

β-End 1–31 showed a remarkable potency to disrupt σ1R-σ2R interactions, with an EC_50_ of approximately 10 fM. The shorter sequences, β-End 1–27 and β-End 1–26, retained only a fraction of β-End 1–31 activity, with EC_50_ values in the low–medium nM range. As previously observed for G protein and σ1/2R binding to σ1R oligomers, ^α^*N*-acetylated β-End 1–31 promoted a contrary effect and thus, the σ1R-σ2R interactions were strengthened (Figure 7A). β-End 1–31, 1–26 and their ^α^*N*-acetylated derivatives also modified σ1R-NR1 C1 associations but followed a pattern contrary to that observed for σ1R-σ2R complexes. While nonacetylated β-End 1–31 and 1–26 promoted σ1R-NR1 C1 associations at pM concentrations, ^α^*N*-acetylated 1–31 and 1–26 also separated σ1Rs from NR1 C1 subunits at pM concentrations. The exceptions were β-End 1–27 and its ^α^*N*-acetylated form, which displayed weak or no effects in this paradigm. It seems that Y27 adds complexity to the activity of these peptides, particularly on σ1R-NR1 C1 associations (Figure 7B). The behavior of β-End 1–31 and N-terminal derivatives on σ1R-BiP complexes was quite similar to that observed for σ1R-σ2R interactions. The nonacetylated peptides 1–31, 1–27, and 1–26 disrupted these complexes with potencies in the pM/nM range, which were much weaker than that of β-End 1–31 at σ1R-σ2R interactions. The ^α^*N*-acetylated derivatives of 1–27 and 1–26 promoted σ1R-BiP complexes with potency comparable to that exhibited for σ1R-σ2R interactions. However, ^α^*N*-acetyl β-End 1–31 was weak at σ1R-BiP associations (Figure 7C).

The σ1R human mutant E102R, which has been implicated in amyotrophic lateral sclerosis (ALS) [57], forms complexes with BiP and NR1 C1 proteins but eludes ligand control of these interactions [58]. Interestingly, this point mutation also prevented β-End 1–31 and ^σ^*N*-acetyl β-End 1–31 from altering σ1R E102Q complexes with σ2R and NR1 C1 proteins (Appendix A). This observation further suggested the binding of these pituitary peptides to σ1Rs. β-End 1–31 and ^σ^*N*-acetyl β-End 1–31 may share a binding site on σ1R, and thus, the capacity of β-End 1–31 to disrupt σ1R-σ2R associations was antagonized by the ^α^*N*-acetylated form. Notably, the potency of ^α^*N*-acetyl β-End 1–31 to counteract the β-End 1–31 effect indicated an affinity toward σ1R in the fM range (Appendix A). Since ^α^*N*-acetyl β-End 1–31 supported σ1R-σ2R associations at higher concentrations than necessary to counteract the β-End 1–31 σ1R-σ2R-disrupting effect, it is possible that high concentrations of this peptide favored the conformation that strengthened these σ1R-σ2R associations.

We also evaluated the activity of β-End C-terminal sequences 27–31, 28–31 and 30–31 on the three interactions with σ1Rs studied herein. The tetrapeptide KKGE and the dipeptide GE promoted σ1R-σ2R associations and reduced σ1R-NR1 C1 interactions (Figure 8A); however, the peptides displayed a weak direct effect on σ1R-BiP complexes (Appendix A). The penptapeptide YKKGE, which is probably derived from the cleavage of β-End 1–31 into β-End 1–26 and β-End 27–31, adds a tyrosine at the N-terminus of KKGE. Again, the presence of Y27 modified the activity of the 28–31 (KKGE) peptide, and YKKGE did the opposite, reduced σ1R-σ2R and increased σ1R-NR1 C1 and σ1R-BiP associations. Because YKKGE and KKGE exhibited identical reducing effects on MOR-mediated β-End 1–31 and morphine analgesia, YKKGE is probably converted in vivo into KKGE by aminopeptidases. Utilizing chimeric sequences of β-End 1–31 shed some additional light on the structure-activity requirements of the physiological peptides when acting on σ1R interactions with σ2Rs and NR1 C1 subunits. Thus, β-End (1–5) + (16–31), 2–31 and 6–31 diminished σ1R-σ2R associations, as did the complete β-End 1–31; however, β-End 22–31 promoted their association (Appendix A), as did the physiological β-End 28–31 (KKGE) and β-End 30–31 (GE) (Figure 8A). It seems that β-End 1–31 binds to σ1R in the σ1R-σ2R heterodimer by its middle 16–22 region, and its C-terminal sequence 22–31, when cleaved, may also bind to σ1R. The binding of the β-End 1–31 middle region promoted the separation of σ1Rs from σ2Rs, whereas the C-terminal region increased their association. Thus, the complete β-End 1–31 sequence mostly exhibited a disrupting effect. This effect of β-End 1–31 and β-End 1–27 on the σ1R-σ2R association was counteracted by β-End 28–31, suggesting that these peptides share a common or at least overlapping binding site on σ1R (Appendix A). Moreover, β-End 1–31 did not need Y1 to bind to σ1R, as β-End 2–31 competed with S1RA and ^α^*N*-acetyl β-End 1–31 for binding to this site (Appendix A). On the other hand, the effect of β-End 1–31 promoting the σ1R-NR1 C1 association was present in 1–26 and 2–31 (Figure 7B and Appendix A), but began to separate the complex after the 1–5 sequence was removed in the 6–31 peptide and persisted in 22–31, 28–31 and 30–31 (Figure 8A and Appendix A). Thus, β-End 1–31 required at least its 2–4 sequence to promote σ1R-NR1 C1 complexes and did not need the last 28–31 residues.

For the sake of comparison, the activity of classical σ1R agonists, PRE084, PPCC, and pregnenolone sulfate, and antagonists, S1RA, BD1047, and progesterone, was also evaluated on the three σ1R interactions. The agonists impaired σ1R associations with σ2R and BiP and increased σ1R-NR1 C1 complexes, whereas the antagonists did the opposite (Figure 8B and Appendix A). The pentapeptides methionine- and leucine-enkephalin, as well as their 2–5 sequences and chimeric ^α^*N*-acetylated derivatives, provided interesting results in these paradigms. While met-enkephalin 1–5 (YGGFM) and 2–5 (GGFM) promoted the sigma receptor heterodimer, leu-enkephalin (YGGFL) and 2–5 (GGFL) disrupted this association (Figure 8C). This suggests that M5 and L5 play a critical role in these opposite effects. Again, the Y1 ^α^*N*-acetylation of met-enkephalin and leu-enkephalin altered the activity of the respective enkephalin toward their opposite effect on σ1R-σ2R interactions, which was disruptive for ^α^*N*-acetyl met-enkephalin and protective for ^α^*N*-acetyl leu-enkephalin. The first determinant, M5 or L5, was modified by Y1 ^α^*N*-acetylation in such a manner that the opposite effect was obtained.

With respect to σ1R-NR1 C1 interactions, removing Y1 or ^α^*N*-acetylation of met-enkephalin resulted in the disruption of these complexes. In contrast, the presence of Y1 in met-enkephalin or merely in β-End 1–4 led to an opposite effect of enhancement. This observation agrees with β-End 1–31 needing at least its 2–4 sequence to promote σ1R-NR1 C1 complexes. The presence of leucine 5 independently of the presence of Y1 or its ^α^*N*-acetylation always conferred a disrupting capacity to the leu-enkephalin derivative (Figure 8D).

The interactions of β-End 1–31 and derivatives with σ1R-NR1 C1 complexes demonstrated that they share a binding site, or at least that their recognition sites overlap in σ1R (Appendix A). Interestingly, the profile of the β-End 1–31 peptides at σ1R-BiP heterodimers revealed that ^α^*N*-acetyl β-End 1–31 is inactive in this ER system. The presence of free Y1 with (1–31) or without the C-terminal sequence (1–26 and 1–27) provided agonist activity to disrupt this interaction. The ^α^*N*-acetylation of Y1 in the derivatives without C-terminal sequences (^α^*N*-acetyl β-End 1–26 and ^α^*N*-acetyl β-End 1–27) resulted in antagonist activity and counteracted agonists’ effects. However, ^α^*N*-acetyl β-End 1–31 and β-End 28–31 neither exhibited activity at σ1R-BiP nor reduced the effect of β-End 1–31 (Appendix A).

## 3. Discussion

^α^*N*-acetyl β-End 1–31 prevented or alleviated a series of negative symptoms in animals, such as the brain injury caused by pMCAO, the convulsive syndrome promoted by icv-injection of NMDA, and mechanical allodynia caused by unilateral sciatic nerve CCI. Notably, these positive effects were prevented by β-End 1–31 and exogenous σ1R agonists and absent in σ1R^−/−^ mice. At the molecular level, σ1R forms oligomers [47], and establishes regulatory interactions with a series of signaling proteins, such as NMDAR NR1 C1 subunits [28] and BiP [37], and we have now demonstrated σ1R interactions with σ2Rs. In this scenario, β-End 1–31 and physiological derivatives influenced σ1R oligomerization as well as the above-indicated associations. The access of these pituitary peptides to σ1Rs in neural membranes was suggested through receptor binding studies. Initially, tritiated naloxone and morphine were the labeled probes, and mouse brain synaptosomes the source of receptors. β-End 1–31 exhibited an affinity to MORs in the low nM range, and ^α^*N*-acetyl β-End 1–31 displayed a very weak MOR affinity in the μM range [45,46]. Later, when [^125^I]-Tyr^27^ β-End 1–31 was available, a very high affinity of β-End 1–31 was detected, and ^α^*N*-acetyl β-End 1–31 displayed high potency to abolish approximately 20–30% of 5 pM [^125^I]-Tyr^27^ β-End 1–31 specific binding, which was probably the σ1R site. Similar competing behavior was observed for ^α^*N*-acetyl β-End 1–27 [14]. 

The binding data may explain why ^α^*N*-acetyl β-End 1–31 abolishes approximately 50 to 60% of β-End 1–31 and morphine analgesia. This antagonism is achieved at remarkable icv fmol doses and does not increase with 100,000-fold higher doses, and the effect of such single low doses is observed for at least 24 h [14]. We have now confirmed these observations and extended the study to practically all physiological derivatives of β-End 1–31. The negative or positive regulatory effects of these peptides on opioid analgesia were absent in σ1R^−/−^ mice. Since σ1R was described as a part of a tonic antiopioid system regulated by σ1R agonists and antagonists [31], β-End 1–31 may be classified as a σ1R agonist, whereas the ^α^*N*-acetyl derivatives and the C-terminal sequences 28–31 and 30–31 are more similar to σ1R antagonists.

These observations reasonably suggest that σ1R participates in the effects of β-End 1–31 and its ^α^*N*-acetyl derivative, and that these pituitary peptides are endogenous regulators of σ1R, which in vivo gain access to MORs in the midbrain gray matter and medulla at fmol icv doses and in vitro binding assays at fM or low pM concentrations. The positive effects of ^α^*N*-acetyl β-End 1–31 on mechanical allodynia, convulsive syndrome and focal cerebral ischemia were attained at higher doses than those modulating opioid analgesia. It is possible that to attain control of such nonphysiological issues, these neuropeptides must reach deep brain structures and recruit a larger number of σ1Rs.

With respect to analgesia, β-End 1–31 binds to and activates MORs and DORs, and within MORs, β-End 1–31 may reach a pool that is regulated by σ1Rs and ^α^*N*-acetyl β-End 1–31. In fact, σ1Rs and MORs colocalize in neuronal plasma membranes and coprecipitate in neural cells [28]; σ1Rs and MORs also coprecipitate in transfected cells [59], and in in vitro studies, their association is promoted by calcium [56]. Early studies proposed a specific receptor for β-End 1–31, the ε receptor [19,20], which exhibits affinity for benzomorphans and is probably related to MOR [34,35]. σ1R, which allosterically regulates GPCRs, such as MOR, also binds to benzomorphans [59]. The physicochemical properties of σ1R and its exogenous ligands are compatible with β-End 1–31 binding to this receptor. σ1R interacts with cholesterol through cholesterol binding domains in its C-terminal sequence [60]. This binding is abolished by σ1R ligands, suggesting that these domains are part of the σ1R drug-binding site. Indeed, σ1R ligands are typically positively charged hydrophobic or amphipathic molecules; thus, these molecules can interact with hydrophobic domains of the receptor [61].

β-End 1–31 is a cationic amphiphilic peptide with a positive 15–31 region that contains its maximum charge at N25-A26 before Y27, which is the point of 28–31 cleavage. The β-End 1–5 sequence (met-enkephalin) is followed by a flexible 8–11 region and an amphiphilic 15–25 helix, which exhibits high hydrophobicity (DNASTAR Lasergene v.17). The hydrophobic domain resulting from the formation of the helical structure covers one-half of the helix surface and, in the α-helical conformation, is continuous and twists along the length of the helical axis [62]. σ1R exhibits amphipathic properties in its calcium-regulated C-terminal region, which interacts with BiP in the ER [63]. The binding of ligands to this region of σ1R may alter its conformation and thus promote changes in its oligomeric organization [47,48], and in its association with other signaling proteins [28]. The σ1R C-terminal sequence contains two steroid binding-like domains (SBDL I and II) [63]. The human mutation E102Q located in the SBDL I (91–109) prevents classical ligands from regulating σ1R interactions with NR1 C1 NMDAR subunits and BiP [58], and our study indicates that ^α^*N*-acetyl β-End 1–31 and β-End 1–31 have also lost their ability to regulate σ1R-σ2R and σ1R-NR1 C1 complexes.

In in vitro assays, β-End 1–31 showed high potency to stabilize σ1R oligomers and disrupt σ1R-σ2R interactions. The latter effect, although weaker, was also exhibited by its shorter sequences 1–26, 1–27, 2–31, 6–31 and the fusion peptide (1–5)–(16–31). However, this activity on the heterodimer was not observed with the C-terminal sequences 22–31, 28–31 and 30–31. Thus, our in vitro observations suggest that β-End 1–31 and ^α^*N*-acetyl β-End 1–31 bind to σ1R through the basic amphiphilic 15–25 helix, while in the whole molecule, the C-terminal sequence probably stabilizes this interaction. This structural analysis of β-End 1–31 compares with that of Blanc et al., which shows that disruption of β-End 1–31 helical structure does not prevent its opioid analgesic effect; thus, modified β-End 1–31 probably acts at MORs and DORs without σ1R regulation, but the neuropeptide is now inactive at the RVD, putative ε receptor [64].

It was suggested that σ1R may regulate MOR transduction [59], and we have now shown that σ1R oligomers bind to inactive G proteins and to σ2Rs influencing MOR signaling. Accordingly, in σ1R^−/−^ mice, the analgesic effects of icv β-End 1–31, morphine and DAMGO significantly increase [28], and we have now shown that in these mice, the association of MORs with diverse classes of G proteins, such as Gi/Go/Gq-11/Gz, is augmented. It is possible that the σ1R negatively regulates MOR activation of G proteins. In fact, the icv administration of purified σ1Rs to σ1R^−/−^ mice diminishes MOR analgesia and restores NMDAR negative regulation of MOR signaling [28]. σ1R can be found in lipid rafts, forming oligomeric structures that may facilitate its relation with diverse signaling proteins, such as ionic channels and GPCRs [65,66]; the regulatory role of σ1Rs on G protein signaling resembles that of the glial integral-membrane protein caveolin, which binds to different classes of G proteins and kinases in their inactive form [67]; however, it is unknown whether the interaction of caveolin with signaling proteins is regulated by ligands.

In ex vivo assays, β-End 1–31 promoted the exchange of σ2Rs by σ1Rs at the σ1R oligomers and increased σ1R-G protein interactions. In contrast, ^α^*N*-acetyl β-End 1–31 exchanged σ1Rs with σ2Rs to release σ1R-bound G proteins in a concentration-dependent fashion. It is therefore possible that σ1R oligomers coordinate with σ2Rs to control the extent and classes of G proteins regulated by GPCRs, such as MOR. In these circumstances, ^α^*N*-acetyl β-End 1–31 and β-End 1–31 may take advantage of this regulation, and their binding to σ1R restrains or promotes certain GPCR-regulated signaling pathways. Therefore, σ1R oligomers bind and stabilize G proteins in their inactive GDP form, but this binding can be regulated by the σ1R ligands studied herein with the collaboration of σ2Rs. The introduction of the ^α^*N*-acetyl in Y1 removes the affinity of β-End 1–5 to the MOR [14] and drastically changed the profile of β-End 1–31 in the σ1R-G protein paradigm. The functional relevance of such a small change (Y1 ^α^*N*-acetylation) on the β-End 1–31 molecule has also been found in its interaction with the amphiphilic tetradecapeptide mastoparan. This peptide mimics ligand-activated GPCRs and stimulates the exchange of GDP by GTP at Gα subunits [68] and thus diminishes MOR signaling. Notably, only ^α^*N*-acetylated forms of β-End 1–31, 1–27 and 1–17 protect MOR analgesia from mastoparan-impaired effects [69].

A subanalgesic icv dose of β-End 1–31 greatly enhanced MOR coprecipitation with σ2Rs and mostly Go/i2 proteins. This observation may account for the reported effects of low doses of β-End 1–31 enhancing morphine and levorphanol analgesia [70]. Because icv analgesic doses of morphine reduce the coprecipitation of Gα subunits with MORs [71], subanalgesic doses of icv β-End 1–31 promoted MOR signaling by a mechanism not related to MOR activation. Thus, β-End 1–31 acting on σ1Rs may enhance MOR signaling capacity and favor certain signaling pathways as well. ^α^*N*-acetyl β-End 1–31 by the icv route moderately increased MOR coprecipitation with σ2R, while ex vivo it reduced the presence of G proteins at σ1R oligomers, and accordingly, MOR regulation of Gi2 proteins barely increased. The higher icv dose of 10 nmol S1RA did not enhance morphine analgesia because this dose augmented MOR coprecipitation with Gi2/o proteins and σ2Rs and thus diminished the regulation of Gz proteins by MOR. Our study suggests that G proteins and σ2Rs compete for σ1Rs binding. In this scenario, β-End 1–31 and its ^α^*N*-acetyl derivative tilt the balance in one direction or another, favoring the union of G proteins or σ2Rs, respectively, to the σ1R oligomers. Indeed, σ1R oligomers captured G proteins in neural tissue from σ2R^−/−^ mice, but ^α^*N*-acetyl β-End 1–31 did not release G proteins or σ1Rs from the σ1R oligomers. σ2R^−/−^ mice exhibited impaired MOR regulation of G proteins, which correlates with their reduced MOR-mediated analgesic responses [54]. Therefore, σ2R was necessary for σ1R oligomers to feed MORs with G proteins and for β-End 1–31 and ^α^*N*-acetyl β-End 1–31 to regulate this mechanism.

With respect to the shorter sequences, β-End C-terminal tetrapeptide 28–31 also disrupted σ1R-G protein associations, although less strongly than by the complete ^α^*N*-acetyl β-End 1–31 peptide. Therefore, the high affinity that the entire sequence exhibited for σ1R possibly resides in the amphiphilic helix. The G proteins that remain associated with σ1R oligomers after ^α^*N*-acetyl β-End 1–31 is administered may also affect the signaling profile of MORs and of other GPCRs, such as α2 drenoceptors, but not DORs or CB1Rs. Thus, ^α^*N*-acetyl β-End 1–31 has relevant implications in the signaling pathways that are triggered by biased agonists at a single GPCR.

Morphine and its derivatives, as well as β-End 1–31, when injected by the icv route, mostly regulate the pertussis toxin- and N-ethyl maleimide-resistant Gz protein and, to a lesser extent, Gi/o proteins to promote supraspinal analgesia [72,73]. Gz proteins are rarely found outside the brain, and their specific regulators RGSZ1 and RGSZ2 also diminish at the spinal cord level [74,75]. Systemic opioids given by the intraperitoneal or subcutaneous route able to cross the BBB mostly reach MORs at the spinal cord level [76], and thus, Gz proteins are less relevant in the production of systemic morphine analgesia [77]. Notably, MOR-binding agonists of pentapeptide structure prefer Gi/o/q-11 proteins rather than Gz proteins [72,78]. Increased regulation of Gi/o/q-11 proteins with a parallel decrease in Gz regulation favors DAMGO signaling and reduces that of morphine and β-End 1–31 at MORs. In fact, an icv injection of ^α^*N*-acetyl β-End 1–31 reduces supraspinal analgesia of β-End 1–31 and morphine but enhances that of DAMGO, in which morphine then antagonizes the activity of DAMGO [15].

The effect of exogenous σ1R agonists and antagonists on morphine and β-End 1–31 analgesia was disparate. While classical σ1R antagonists enhanced morphine analgesia, they reduced that of β-End 1–31, and no significant effects were produced by the agonists in this paradigm. Our data showed that σ1R antagonists, such as S1RA, bind to σ1Rs and release Gi/o proteins. As a result, these compounds impaired the analgesic activity of the σ1R agonist β-End 1–31 by competing with its binding to σ1Rs and by their opposite effects on MOR-regulated G proteins. In the case of morphine, we have previously shown [28], and confirmed in this study, that σ1R antagonists, such as S1RA, given by the icv route exhibit a biphasic effect with doses enhancing opioid analgesia, and higher doses barely producing this effect or even diminishing morphine analgesia. Upon the activation of MORs, exogenous σ1R antagonists first uncouple σ1Rs from NMDARs, thus reducing the impact of glutamate NMDAR negative feedback on MOR signaling and increasing morphine supraspinal analgesic effects [28]. This was not observed for β-End 1–31 because its binding to σ1Rs reduced the activating access of σ1Rs to NMDARs. At higher icv doses, classical σ1R antagonists increased the regulation of Gi2/o proteins by MOR and thus diminished the signaling capacity of icv morphine at MOR-Gz complexes. The σ1R agonists did not release G proteins from σ1R oligomers but prevented σ1R antagonists and ^α^*N*-acetyl β-End 1–31 from reducing β-End 1–31 analgesia. When binding to σ1Rs, β-End 1–26, 1–27 and ^α^*N*-acetylated derivatives did not reproduce the effects of β-End 1–31 on σ1/2R and G protein association with the MORs and then impaired β-End 1–31 analgesia [12]. Therefore, β-End 1–31 acted as a σ1R agonist that binds to MORs and promotes opioid analgesic effects, and ^α^*N*-acetyl β-End 1–31 and classical σ1R antagonists, but not agonists, reduced the enhancing effects of β-End 1–31 on MOR G protein signaling and diminished β-End 1–31 analgesia.

A similar reasoning may explain the antagonism of β-End 28–31 on β-End 1–31, and morphine analgesia and its prevention by classical σ1R agonists. The β-End 28–31 may have functional relevance because the pituitary contains high levels of its complementary peptides β-End 1–27 and ^α^*N*-acetyl β-End 1–27 [9]. In fact, the 30–31 dipeptide, which lacks affinity toward MORs, regulates certain morphine effects, such as tolerance, dependence and withdrawal [18]. The classical σ1R antagonists also restore morphine analgesia in tolerant mice [32]. Notably, a single icv pmol dose of ^α^*N*-acetyl β-End 1–31 significantly alleviates the morphine withdrawal syndrome precipitated by naloxone in rats and mice [17]. The development of tolerance/dependence to chronic morphine and the incidence of the withdrawal syndrome can be attenuated by reducing the availability of certain classes of brain G proteins, mostly Gz and Gi2 [79]. Most likely, the ^α^*N*-acetyl β-End 1–31 that acts through σ1Rs switches MOR signaling from morphine-tolerant signaling pathways to others that exhibit low tolerance/dependence to the opioid. The possibility of the 30–31 dipeptide acting through σ1Rs in these paradigms merits consideration.

The enkephalins also provided valuable information on the regulatory role of σ1Rs on MOR function. In this respect, leu-enkephalin and met-enkephalin exhibited differences at σ1Rs. Met-enkephalin and its aminopeptidase 2–5 product promoted σ1R-σ2R associations, while Leu-enkephalin and its 2–5 metabolite diminished these associations. In this scenario, leu-enkephalin behaved as β-End 1–31 while, met-enkephalin as ^α^*N*-acetyl β-End 1–31. Leu-enkephalin 1–5 and 2–5 products, dissociated σ1R-NR1 C1 complexes (uncoupling MOR from NMDAR negative feedback), and met-enkephalin promoted this association. It is therefore possible that these differences account for the reported enhancing effects of low doses of leu-enkephalin on morphine analgesia [80], and for the absence or even diminishing effect of met-enkephalin on MOR analgesia [70,81]. The effect of met-enkephalin is not observed for β-End 1–31, probably because in the whole 1–31 molecule, the amphiphilic helix and not the 1–5 sequence binds to σ1R, and the 1–31 C-terminal sequence has already uncoupled MOR from the influence of NMDAR (with σ1R-NR1 C1 as a σ1R antagonist). Thus, the 1–5 sequence YGGFM within β-End 1–31 mostly binds to opioid receptors, or when Y1 is ^α^*N*-acetylated, the sequence contributes to the release of σ1R-bound G proteins. Since met- and leu-enkephalins exhibited opposite effects in the abovementioned paradigm, the C-terminal methionine or leucine determines the regulatory effects on morphine analgesia.

It is highly probable that σ1R participates in the pharmacologically defined ε receptor, which was initially described in the RVD as specific for β-End 1–31 [19]. The structural requisites for β-End 1–31 to interact with this RVD receptor are distinct from those needed to activate guinea pig ileum MORs. While the N-terminal met-enkephalin segment is necessary to activate both receptors, the amphiphilic helix in residues 13–23 and an aromatic residue in 18 β-End 1–31 determine its activity at the RVD [20,62,64,82]. Moreover, after blocking access to MOR and DOR, β-End 1–31 still produces effects in the RVD, which are sensitive to naloxone [83]. 

The ε receptor was also described in the CNS and was mostly related to β-End 1–31 and analgesia [19,23]. The development of tolerance to a specific opioid receptor in isolated organs and the limited analgesic cross-tolerance between morphine and β-End 1–31 in mice and rats supports the idea of a receptor that is mostly acted upon by β-End 1–31 [22,84,85]. However, other studies challenge this idea, and morphine does behave as a partial agonist in the RVD perfused with low calcium, which may reveal that RVD contains a low number of MORs. In this case, full agonists would efficiently inhibit electrically induced twitching but not morphine, which displays low intrinsic activity [86]. Notwithstanding, the low abundance of MORs in the RVD does not contradict the idea that a σ1R-MOR complex is the ε receptor because their association is highly dependent on calcium; thus, with low calcium, the MOR may exist as mostly uncoupled from σ1Rs [28,56], increasing morphine activity. In agreement with the participation of σ1Rs and MORs in the ε receptor, in triple MOR, DOR, and KOR ko mice β-End 1–31 does not promote GTPγS binding to Gα subunits [25], and in σ1R^−/−^ mice ^α^*N*-acetyl β-End 1–31 did not alter MOR analgesia. Thus, morphine and β-End 1–31 promote analgesia by mostly binding to MORs and MOR-σ1R receptors, and the fraction of β-End 1–31 and morphine analgesia abolished by ^α^*N*-acetyl β-End 1–31 may be related to σ1R-regulated MOR, the ε receptor. The important aspect is the different activity of morphine at the putative ε receptor in RVD vs. that in CNS; partial agonist or antagonist in the former and analgesic in the latter. As we mentioned before, the Gz protein is essential for MOR agonists, such as morphine and its derivatives, to promote analgesia when injected by the icv route [73], but this pertussis-toxin-resistant G protein is rarely found outside the CNS [75]. Thus, in the presence of a reduced number of MORs and the absence of Gz proteins, morphine behaves as a partial agonist when activating Gi/o proteins.

In summary, the data obtained are compatible with the binding of β-endorphin 1–31 and ^α^*N*-acetylated derivative to the σ1R. The in vivo effects of these peptides were absent in σ1R^−/−^ mice. These endogenous peptides altered, in a concentration-dependent fashion, the associations of σ1Rs with partner proteins such as σ2R, NR1 C1 and BiP, and these effects were shared by typical ligands of σ1Rs. β-endorphin 1–31, ^α^*N*-acetyl β-endorphin 1–31 and physiological derivatives competed with typical σ1R ligands for their effects in in vivo and in vitro paradigms. The regulatory effects of ^α^*N*-acetyl β-endorphin 1–31 on MOR supraspinal analgesia were achieved at low fmols, and the disrupting effects of β-endorphin 1–31 on σ1R-σ2R and σ1R-BiP associations were observed at low fM and pM concentrations, respectively. Similarly, ^α^*N*-acetyl β-endorphin 1–31 disrupted σ1R-NR1 C1 complexes at pM concentrations. The associations established between the ALS-related σ1R human mutant E102Q with partner proteins eluded regulation by classical σ1R ligands, but also by β-endorphin 1–31 and ^α^*N*-acetyl β-endorphin 1–31. 

We believe that this study has unveiled a new role for the pituitary circulating hormones β-End 1–31 and its ^α^*N*-acetylated derivative as regulators of GPCR signaling through their binding to σ1R. As GPCRs are responsible for the proper conduction of many physiological processes, such as intercellular communication, neuronal transmission, and hormonal signaling, and are involved in many pathological processes, such as neuropsychiatric, neurodegenerative, and vascular disorders and vulnerability to drugs, our findings open a new perspective for the study of disparate GPCR dysfunctions.

## 4. Materials and Methods

### 4.1. Animals and Drugs

Wild-type male albino CD1 mice, controls for the homozygous CD1 male sigma 1 receptor (σ1R^−/−^) knockout mice (ENVIGO, Milano, Italy), were used in this study. All mouse housing, breeding and experimental protocols were performed in strict accordance with the guidelines of the European Community for the Care and Use of Laboratory Animals (Council Directive 2010/63/EU) and Spanish law (RD53/2013) regulating animal research. The use of drugs, experimental design and sample-size determination were approved by the Ethical Committee for Research of the CSIC (CAM PROEX 317/16). Further details can be found in our previous publications [54]. The mice were used when they were about 8 weeks of age, and the number of animals used in this study were: CD1 wild-type, 300, CD1 σ1R^−/−^, 145.

The compounds used in this study were: morphine sulfate (MOR agonist, Merck, Darmstadt, Germany); β-End 1–31 human (#RP11344, GenScript, Piscataway, NJ, USA); S1RA (#16279, Cayman Chemical, Ann Arbor, MI, USA); DAMGO (#1171, Tocris Bioscience, Bristol, UK); DPDPE (#1431, Tocris); WIN55,212–2 (#1038, Tocris); NMDA (#0114, Tocris); BD1047 (#0956, Tocris), BD1063 (#0883, Tocris); (±)-PPCC oxalate (#3870, Tocris); PRE084 (#0589, Tocris); 4-IBP (#0748, Tocris); Clonidine hydrochloride (#0690, Tocris); [D-Ala2]-Deltorphin II (#1180, Tocris); Progesterone (#P7556, Merck-Sigma Madrid, Spain); Pregnenolone sulfate (#P162, Merck-Sigma). The following peptides: β-End 1–31; ^α^*N*-acetyl β-End 1–31; β-End 1–26; β-End 1–27; ^α^*N*-acetyl β-End 1–26; ^α^*N*-acetyl β-End 1–27; β-End 27–31; β-End 28–31; β-End 30–31 (PepMic Co. Suzhoo, China); the β-End 1–16; β-End 1–4; β-End 2–31; β-End 6–31; β-End (1–5) + (16–31); β-End 22–31; ^α^*N*-acetyl β-End 1–16; Leucine-enkephalin 1–5, 2–5 and ^α^*N*-acetyl 1–5; Methionine-enkephalin 1–5, 2–5 and ^α^*N*-acetyl 1–5 were custom-synthesized (purity ≥ 98%) by GenScript. Doses and treatment intervals were selected based on previous studies and pilot assays. Test drugs were dissolved in saline, except PPCC and pregnenolone sulfate, which were prepared in a 1:1:18 (v/v/v) mixture of ethanol:Kolliphor EL (#C5135, Merck-Sigma):physiological saline. The compounds were injected intracerebroventricularly (icv) in 4 µL as described [28,87].

### 4.2. Permanent Unilateral Middle Cerebral Artery Occlusion (pMCAO) and the Determination of Infarct Size

Focal cerebral ischemia was induced via pMCAO, as described previously [39]. Briefly, mice were anesthetized and a vertical skin incision was made between the left eye and ear under a dissection microscope. After drilling a small hole in the cranium at the level of the distal portion of the middle cerebral artery, the artery was occluded by cauterization. Flow obstruction was visually verified. Animals showing subdural haemorrhages or signs of incorrect surgery were immediately excluded from the study (<5% in each group). The mice were returned to their cages after surgery, kept at room temperature, and allowed food and water ad libitum. Strong lesion reproducibility was observed. We excluded mice from further studies if excessive bleeding occurred during surgery, mice failed to recover from anesthesia within 15 min, or hemorrhage was found in the brain during post-mortem examination. The investigator performing the pMCAO surgery was blinded to the treatment group. To determine the infarct size 48 h after surgery, animals were euthanized and their brains were removed, after which six 1 mm thick coronal brain slices (Brain Matrix, WPI, UK) were obtained. The sections were stained with 2,3,5-triphenyltetrazolium chloride (1% TTC, Merck-Sigma). Samples were taken from each side of all brain sections with a digital camera (Nikon Coolpix 990, Tokyo, Japan), and the extent of unstained infarct area (expressed in mm^2^) was integrated from the total area as an orthogonal projection.

### 4.3. NMDA-Induced Seizures

Seizures were induced by injection of NMDA (1 nmol/mouse icv, in a volume of 4 μL sterile saline) as described by others [41]. The dose of 1 nmol NMDA was selected as the minimal dose that reliably induced the appearance of tonic seizures in at least 80% of treated mice. Immediately after injection, animals were placed in a transparent box (20 × 20 × 30 cm) and were observed for a period of 3 min. The seizure activity consisted of a mild myoclonic phase (immobility, mouth and facial movements, tail extension, circling), rearing (violent movements of the whole body, rearing), wild running (episodes of running with explosive jumps), clonic convulsions (characterized by rigidity of the whole body, including limb flexion/extension), followed by continuous/repetitive seizure activity (tonic seizures) and, in approximately 15–20% of the animals, death. The episode typically began a few seconds after injection and evolved to its maximal intensity in less than 1 min. The results are expressed as the percentage of mice exhibiting the aforementioned signs and the mean latencies of the first body clonus.

### 4.4. Nerve Injury Pain Model

After the basal mechanical sensitivity of the mice was tested, neuropathic pain was induced by chronic sciatic nerve constriction injury (CCI) surgery under isoflurane/oxygen anesthesia [44] using the procedure described by Bennett and Xie [43] with modifications. Briefly, a 0.5 cm incision was made in the right midthigh, the biceps femoris muscle was separated, and the sciatic nerve was exposed proximal to its trifurcation. Two ligatures (5/0 braided silk suture; Lorca Marin, Murcia, Spain, 70,014) were tied around this nerve approximately 1 mm apart until a short flick of the ipsilateral hind limb was observed. The incision was then closed in layers with a 4–0 Ethicon silk suture. The same procedure was used for sham surgery, except that the sciatic nerve was exposed but not ligated. The tactile pain threshold of both the ipsilateral and contralateral hind paws was then assessed before and on different days post-surgery. The mice were individually placed in a transparent plastic cage with a wire-mesh bottom that allowed full access to the paws. After a habituation period of 20 min, a mechanical stimulus was delivered to the plantar surface from below the floor of the test chamber to measure allodynia using an automatic von Frey apparatus (Ugo Basile #37,450, Comerio, Italy). A steel rod (0.5 mm diameter) was pushed against the hind paw over a 10 s period as the force increased from 0 to 10 g. When the mouse withdrew its hind paw, the mechanical stimulus was automatically stopped, and the force at which withdrawal occurred was recorded. At each time point, three separate threshold measurements were obtained from each hind paw and then averaged.

### 4.5. Evaluation of Analgesia

The response of the animals to nociceptive stimuli was determined by the warm water (52 °C) tail-flick test as previously described [28,54]. In this tail-flick analgesic test, a thermal noxious stimulus is applied to promote flicking of the mouse’s tail and opioids given icv increase the time elapsed between application of the stimulus and the flick. This response involves a spinal reflex that is facilitated by the brain stem nociceptive modulating network. Baseline latencies ranged from 1.6 to 2.1 s. A cut-off time of 10 s was used to minimize the risk of tissue damage. Drugs were icv-injected into the lateral ventricles in a volume of 4 μL, and antinociception was assessed at different time intervals thereafter. Saline was likewise administered as a control. Antinociception is expressed as a percentage of the maximum possible effect (MPE = 100 × [test latency-baseline latency]/[cut-off time (10 s) − baseline latency]).

### 4.6. Immunoprecipitation and Western Blotting

The selectivity of the MOR antibody used and the reliability of the immunoprecipitation procedure has already been reported [71,74,88]. In immunoprecipitation studies, the cerebral cortices or periaqueductal grey matter (PAG) from 4–6 mice are typically pooled, as described previously [89]. The affinity-purified IgGs against the MOR second extracellular loop (2EL) (205–216: MATTKYRQGSID; GenScript Co) were labeled with biotin (#21217, ThermoFisher Scientific, Madrid, Spain). The Nonidet P-40 solubilized proteins were incubated overnight at 4 °C with biotin-conjugated primary antibodies raised against the target protein, the immunoprecipitates were recovered and resolved by sodium dodecyl sulfate (SDS)-poliacrylamide gel electrophoresis (PAGE) on 4–12% Bis-Tris gels (#NP0341, Invitrogen, Thermo Fisher Scientific, Madrid, Spain), with MES SDS as the running buffer (#NP0002, Invitrogen). The separated proteins were then transferred onto 0.2 µm polyvinylidene difluoride (PVDF) membranes (#162–0176, Bio-Rad, Madrid, Spain). The membranes were probed overnight at 6 °C with the selected primary antibodies diluted in Tris-buffered saline [pH 7.6; TBS] + 0.05% Tween 20 (TTBS), detecting antibody binding with secondary antibodies conjugated to horseradish peroxidase. The images of the Western blots and the antibody binding were visualized by chemiluminescence (#170–5061, Bio-Rad) and recorded on an ImageQuantTM LAS 500 apparatus (GE Healthcare, Chicago, IL, USA) typically selecting the area containing the target protein in each blot. The software automatically calculates the optimal exposure time for each of the areas specified to provide the strongest possible signal for accurate quantification of the sample. Protein immunosignals were measured using the area of the strongest signal for each group of samples studied (average optical density of the pixels within the object area/mm^2^; AlphaEase FC software); the grey values of the means were then normalized within the 8 bit/256 grey levels [(256 − computed value)/computed value]. In the immunoprecipitation studies, the secondary antibodies were directed to either the heavy or light IgG chains of the primary antibodies as needed [goat anti-rabbit heavy chain HRP-linked antibody (#7074s, Cell Signalling, Werfen, Barcelona, Spain) and mouse anti-rabbit light chain antibody (MAB201, Merck-Sigma)], and thus provided a control for the gel loading of the samples [90].

The primary antibodies used in Western blotting were: anti-MOR second extracellular loop (2EL) aa 208–216 [91], anti-MOR Ct aa 387–398 (GenScript Co.), anti-Gαi2 and anti-Gαz [72,92], anti-Gαo (GC/2, NEI-804 Du Pont-New England Nuclear Research Products, Boston, MA) [49], anti-Gαq-11(QL, NEI-809, Du Pont-New England Nuclear Research Products) [49], anti-Gβ1,2 (R. Schulz, University of Munich, Germany), anti-σ2R Nt (#MBS8557626, My BioSource, San Diego, CA, USA), anti-σ1R (#42–3300, Invitrogen), NMDAR NR1 C1 (#AB5046, Merck-Sigma).

### 4.7. Recombinant Protein Expression

The coding region of the full-length murine σ1R (AF004927), its mutated sequence E102Q, σ2R (NM_133706.2) and the cytosolic C0-C1-C2 regions of the glutamate NMDAR NR1 subunit (NM_008169: residues 834–938) were amplified by reverse transcriptase (RT)-polymerase chain reaction (PCR) using total RNA isolated from the mouse brain as the template. 

Specific primers containing an upstream Sgf I restriction site and a downstream Pme I restriction site were used as described previously [28]. The RT-PCR products were cloned downstream of the GST/HaloTag^®^ coding sequence (Flexi^®^ Vector, Promega, Madrid, Spain) and the TEV protease site, and when sequenced, the proteins were identical to the GenBank™ sequences. The vector was introduced into the *E. coli* BL21 (KRX #L3002, Promega) and clones were selected on solid medium containing ampicillin. After 3 h of induction at room temperature (RT), in the presence of 1 mM isopropyl β-D-1-thiogalactopyranoside (IPTG) and 0.1% Rhamnose, the cells were collected by centrifugation and maintained at −80 °C. The fusion proteins were purified under native conditions on GStrap FF columns (#17–5130-01, GE Healthcare) or with HaloLink Resin (#G1915, Promega). When necessary, the fusion proteins retained were cleaved on the column with ProTEV protease (#V605A, Promega) and further purification was achieved by high-resolution ion exchange (#780–0001Enrich Q, BioRad) and electroelution (GE 200; Hoefer Scientific Instruments, San Francisco, CA) (Appendix A). Sequences were confirmed by automated capillary sequencing. Recombinant BiP (#ab78432) was purchased at Abcam (Cambridge, UK).

### 4.8. In Vitro Interactions between Recombinant Proteins: Effect of Ligands on the σ1R-Proteins Interactions

The recombinant σ1R (100 nM) was incubated either with Haloprotein (negative control) or together with the immobilized σ2R, NR1 C0-C1-C2 or BiP proteins in 300 µL of a buffer containing 50 mM Tris-HCl (pH 7.4) and 0.2% CHAPS in the presence of 2.5 mM CaCl_2_ for 30 min at RT. After removal of the unbound σ1R, the agarose-attached complexes were incubated for 30 min with rotation at RT in the presence of increasing concentrations of the ligands in a final reaction volume of 300 µL [50 mM Tris-HCl (pH 7.4), 2.5 mM CaCl_2_ and 0.2% CHAPS]. Ligands were usually dissolved in aqueous solutions; however, when an organic solvent such as DMSO was required to incorporate, e.g., pregnenolone sulfate, the solvent remained below 1% in the assay buffer. Agarose pellets containing the bound proteins were obtained by centrifugation, washed and resuspended thrice in the presence of 2.5 mM CaCl_2_, solubilized in 2× Laemmli buffer plus β-mercaptoethanol, and analyzed by Western blotting.

The detached proteins recovered in the aforementioned procedures were resolved with SDS-PAGE, transferred onto PVDF membranes and proved overnight with primary antibodies as described above. Because all the assays were performed with recombinant proteins, the immune signals provided a single band of the expected size, which was used for the subsequent densitometry analysis. Accordingly, no other regions of the blots provided information and were routinely excluded from the analysis. 

### 4.9. Fishing Assays Using Agarose-σ1R as Bait and Solubilized Mouse Brain Synaptosomes: Effect of Opioid Peptides

For fishing studies, solubilization of neural synaptosomes from cerebral cortices or PAG was carried out at 4 °C in RIPA buffer. The membranes were incubated at 4 °C for 16 h with agitation, and then centrifuged at 10,000× *g* for 5 min. The supernatant was cleared with 20 μL of Sepharose 4B (#17-0120-01, GE Healthcare) pre-equilibrated for 1 h at 4 °C, followed by centrifugation at 4300× *g* for 5 min. The σ1R was immobilized through covalent attachment to N-Hydroxysuccinimide (NHS)-activated Sepharose 4 fast flow (4FF, #17-0906-01; GE) according to the manufacturer’s instructions. 

The solubilized material was incubated for 2 h with either Sepharose 4B (negative control) and agarose-σ1R oligomers in the absence and presence of increasing concentrations of the ligands in 300 µL of a buffer containing 50 mM Tris-HCl (pH 7.5), 50 mM NaCl. Agarose pellets containing the bound proteins were obtained by centrifugation, washed and resuspended four times, solubilized in 2× Laemmli buffer plus β-mercaptoethanol, separated on a 4–12% Bis-Tris gels (#NP0341, Invitrogen), and the G protein subunits, σ1R, and σ2R immunosignals were probed in Western blots.

### 4.10. Cell Culture

CHO cells were cultured in modified Eagle’s medium/nutrient mixture F-12 (DMEM/F12) supplemented with GlutaMax (#31331028, ThermoFisher, Spain), 10% foetal bovine serum (#10500064, ThermoFisher), 100 IU/mL penicillin and 100 μg/mL streptomycin (#15140148, ThermoFisher) at 37 °C in a humidified 5% CO_2_ incubator as described previously [89]. Full-length sigma receptors were subcloned in frame into the pSF-CMV-Puro-NH_2_-CMyc N-terminal tag (σ1R, #OGS3214, Merck-Sigma) and pSF-CMV-Puro-NH_2_-HA N-terminal tag (σ2R, #OGS1500, Merck-Sigma) mammalian plasmids using standard cloning strategies. CHO cells were seeded on 12-well plates at a concentration of 0.35 × 10^6^ cells per well, grown in Opti-MEM™ I Reduced Serum Medium (#31985070, ThermoFisher) to 70–80% confluence, transfected using Lipofectamine 2000 (#11668030, Invitrogen) and incubated for 24 h prior to testing for transgene expression. For all experiments, the culture medium was replaced with serum-free medium before the addition of sigma ligands and incubated for 30 min. Cells were then harvested and cellular extracts were prepared for immunoprecipitation analysis. Receptors were detected by Western blot analysis using an antibody that recognizes either tag: Myc-tag (#MA1 21316 BTIN, Invitrogen), HA-tag (#26183-BTIN, Invitrogen). 

In other experiments, for immunocytochemistry detection, cells were seeded in 8-well glass bottom plates (#154941, ThermoFisher), transfected and fixed with ice-cold methanol for 10–20 min at 4 °C. Sigma receptors were labelled with c-Myc Monoclonal Antibody (9E10), Alexa Fluor 488 (#MA1–980-A488, ThermoFisher) or HA Tag Monoclonal Antibody (16B12), Alexa Fluor 594 (#A21288, ThermoFisher) and visualized by confocal microscopy using a Leica DMIII 6000 CS confocal fluorescence microscope (Leica Microsystems GmbH, Wetzlar, Germany) equipped with a TCS SP5 scanning laser. Two-dimensional images of 1024 × 1024 pixels were acquired with an emission wavelength of 488 or 561 nm using a HCX PL APO CS 40.0 × 1.25 oil immersion objective.

### 4.11. Statistical Analyses

The signals from the Western blot were expressed as the change relative to the controls, which were assigned an arbitrary value of 1. Statistical analyses were performed using the Sigmaplot/SigmaStat v.14.5 package (statistical package for the social sciences (SPSS) Science Software, Erkrath, Germany), and the level of significance (α) was defined as *p* < 0.05. The data were analyzed using one-way ANOVA followed by the Holm–Sidak multiple comparisons test. The power (1-β) of the tests performed at α = 0.05 was always 0.80 (80%).

## Figures and Tables

**Figure 1 ijms-24-00582-f001:**
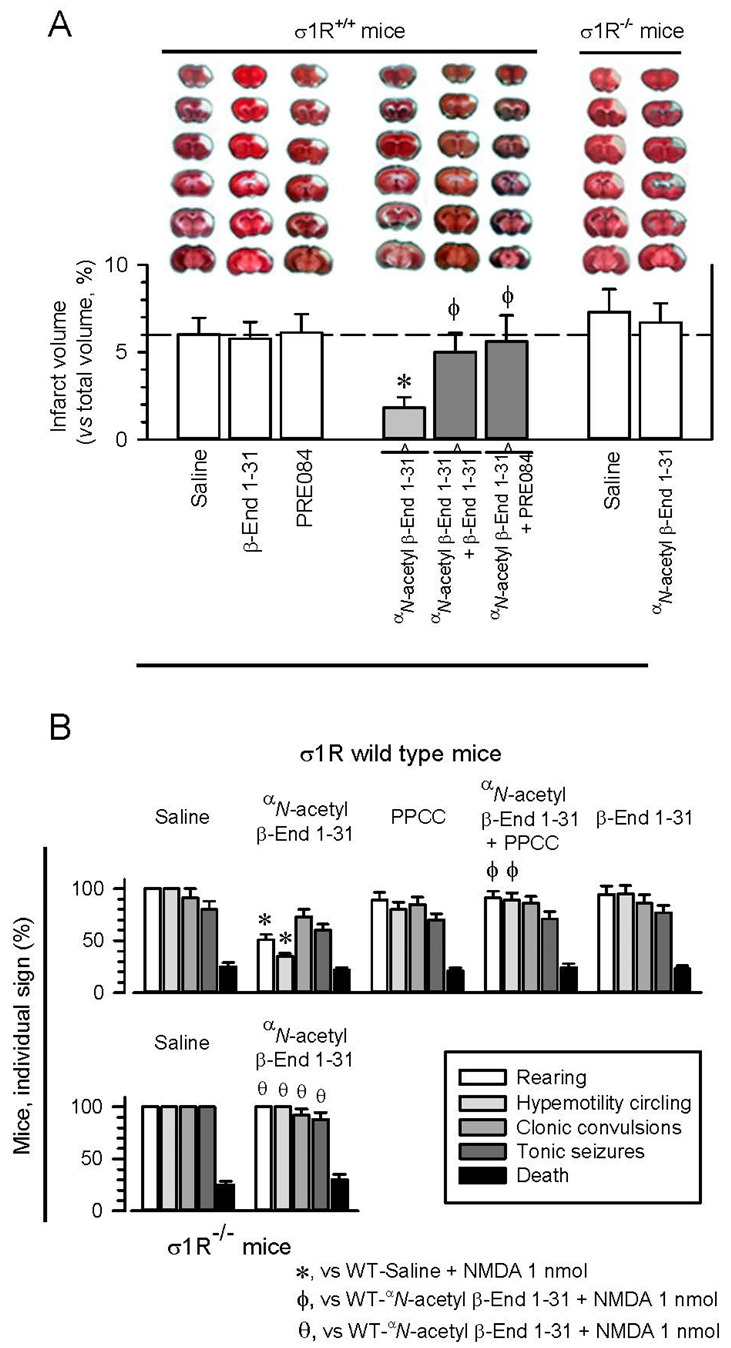
^α^*N*-acetyl β-End 1–31 reduced ischemic brain damage and diminished seizures induced by NMDAR overactivation. Implication of σ1Rs. (**A**) ^α^*N*-acetyl β-End 1–31 administration diminished ischemic brain damage in wild-type but not σ1R^−/−^ mice. Upper panel: representative 2,3,5-triphenyltetrazolium chloride (TTC)-stained brain section images obtained from saline- and drug-treated mice (3 nmol, 1 h after surgery) 48 h after pMCAO. White indicates infarction; red staining indicates normal tissue. Lower panel: the bar graphs quantitatively compare the infarct volume based on TTC staining from wild-type mice treated with saline, β-End 1–31 and the σ1R agonist PRE084 (white bars). The reducing effect of ^α^*N*-acetyl β-End 1–31 was counteracted by β-End 1–31 and PRE084 (gray bars). In σ1R^−/−^ mice, the positive effect of ^α^*N*-acetyl β-End 1–31 was not observed. The groups consisted of five to eight mice, and the data are presented as the mean ± SD. * Significantly different from the saline-treated mice. ϕ Significantly different from mice receiving only ^α^*N*-acetyl β-End 1–31; ANOVA followed by the Holm–Sidak multiple comparisons test, *p* < 0.05, 1-β > 0.80. (**B**) Anticonvulsant effects of ^α^*N*-acetyl β-End 1–31 in a mouse model of seizures induced by NMDAR overactivation. Effects of PPCC, β-End 1–31 and ^α^*N*-acetyl β-End 1–31 on seizures induced by NMDA in wild-type and σ1R^−/−^ mice. The mice received icv 1 nmol of the NMDAR agonist NMDA 30 min after the other drugs (3 nmol) and were then immediately evaluated. Each bar indicates the percentage of mice showing the indicated sign and represents the mean ± SD of eight mice. * Significant difference from the control group that received NMDA and saline instead of the drugs. ϕ Significant difference from the corresponding NMDA-induced behavioral signs exhibited by the group that received only ^α^*N*-acetyl β-End 1–31. θ Significant difference from the corresponding NMDA-induced behavioral signs exhibited by the wild-type group that received only ^α^*N*-acetyl β-End 1–31. ANOVA followed by the Holm–Sidak multiple comparisons test, *p* < 0.05, 1-β > 0.80.

**Figure 2 ijms-24-00582-f002:**
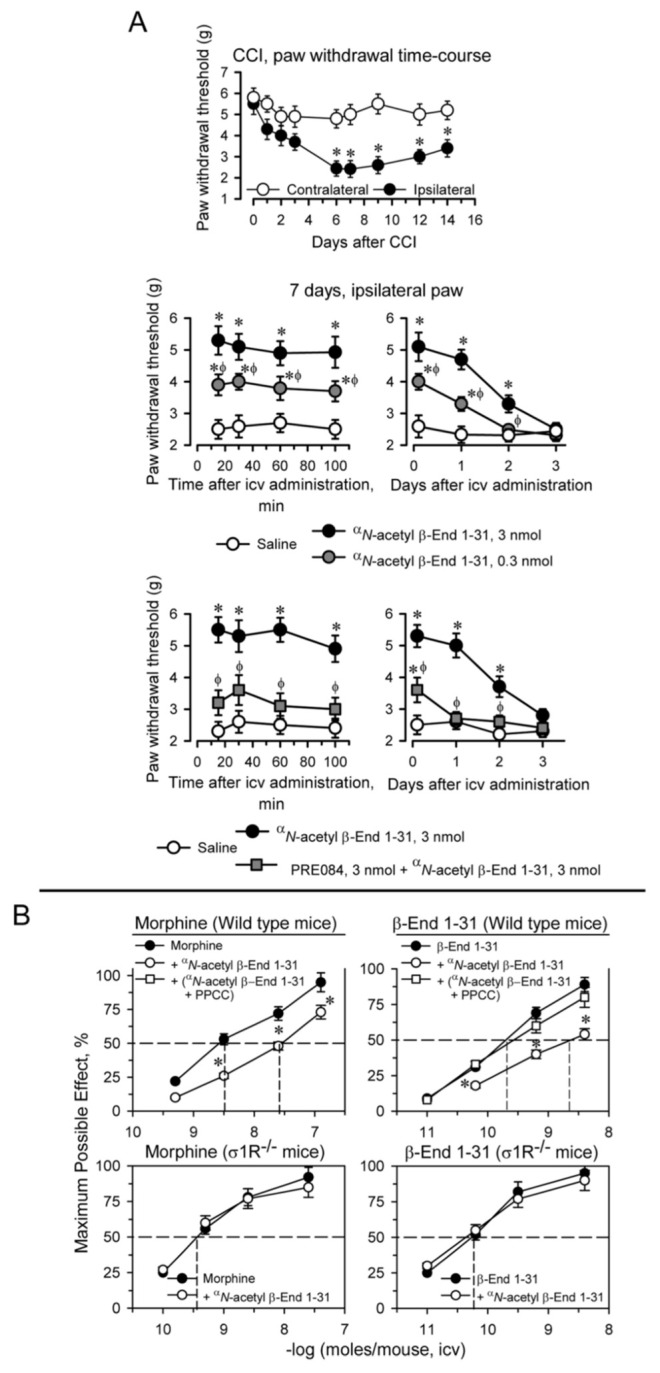
^α^*N*-acetyl β-End 1–31 exhibited antiallodynic effects and reduced β-End 1–31 and morphine analgesia. Implication of σ1Rs. (**A**) Effect of ^α^*N*-acetyl β-End 1–31 on the mechanical allodynia induced by chronic constriction injury (CCI) of the sciatic nerve. Upper panel: the withdrawal thresholds of the contralateral and ipsilateral paws were measured before (indicated as 0) and up to 14 days after surgery. The force (in grams) at which the mice withdrew their paws in response to von Frey hair stimulation was determined as an index of mechanical allodynia. * Significantly different from the value of the contralateral paw. Middle panel: 0.3 and 3 nmol of ^α^*N*-acetyl β-End 1–31 were administered icv 7 days after surgery, and the nociceptive threshold was evaluated at the indicated postinjection intervals (in minutes). For every time interval studied in min or days, * indicates a significant difference from the group that received saline instead of ^α^*N*-acetyl β-End 1–31; ϕ significant difference from the effect of 3 nmol ^α^*N*-acetyl β-End 1–31. Lower panel: the antiallodynic effect of 3 nmol ^α^*N*-acetyl β-End 1–31 was diminished by coadministration of 3 nmol PRE084. * Significantly different from the group which received saline instead of ^α^*N*-acetyl β-End 1–31. ϕ Significantly different from the group that received only ^α^*N*-acetyl β-End 1–31. All data are presented as the mean ± SD of six mice. ANOVA followed by the Holm–Sidak multiple comparisons test, *p* < 0.05, 1-β > 0.80. (**B**) Effect of ^α^*N*-acetyl β-End 1–31 on morphine and β-End 1–31 analgesia in mice. Upper panel: wild-type mice were injected icv with saline, ^α^*N*-acetyl β-End 1–31 (1 nmol/mouse) or the combination of the σ1R agonist PPCC (3 nmol/mouse) plus ^α^*N*-acetyl β-End 1–31 20 min before morphine and β-End 1–31. Analgesia was evaluated by the warm water (52 °C) tail-flick test at the interval postinjection corresponding to their peak effect, 30 min. Lower panel: σ1R^−/−^ mice received 1 nmol ^α^*N*-acetyl β-End 1–31, 20 min before morphine and β-End 1–31 and analgesia was evaluated as described above. Eight mice were used per strain and treatment. * Significant difference with respect to the group treated with morphine or β-End 1–31 alone. ANOVA followed by the Holm–Sidak multiple comparisons test, *p* < 0.05, 1-β > 0.80.

**Figure 3 ijms-24-00582-f003:**
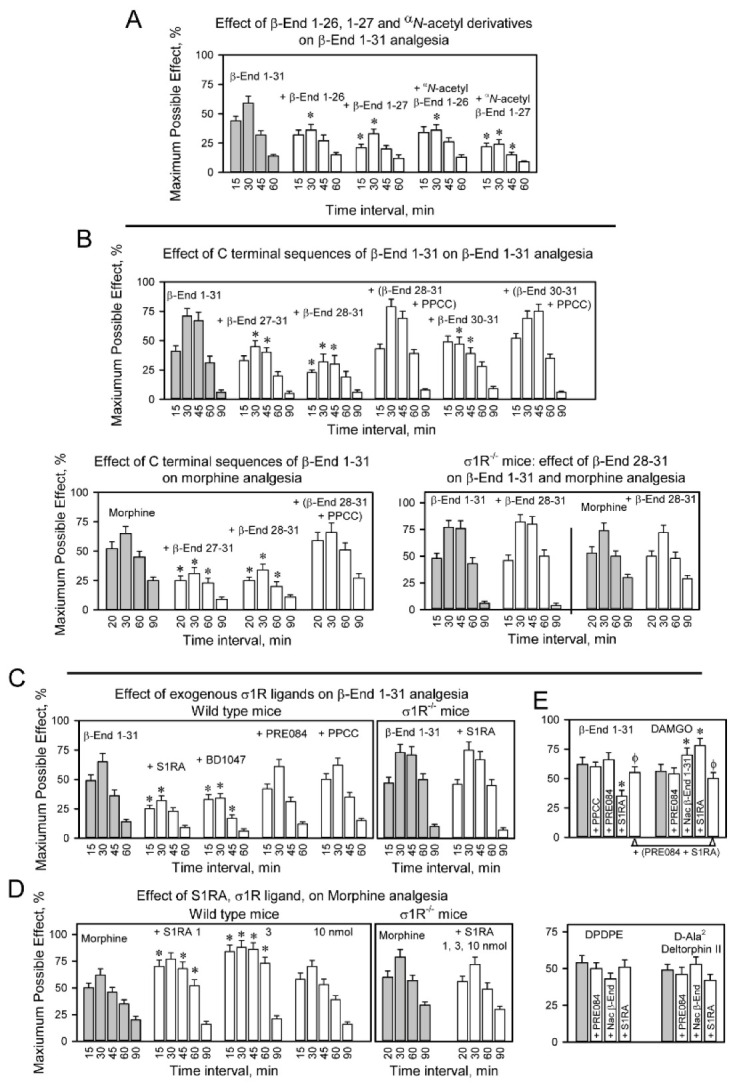
Effect of physiological β-End 1–31 derivatives on morphine and β-End 1–31 analgesia. Implication of σ1Rs. (**A**) The mice were icv-injected with 25 nmol of β-End 1–26, β-End 1–27 and ^α^*N*-acetylated derivatives 20 min before icv 0.5 nmol β-End 1–31. Analgesia was determined at the indicated postinjection intervals. (**B**) CD1 wild-type and σ1R^−/−^ mice were administered C terminal peptide sequences 27–31, 28–31 and 30–31 of β-End 1–31 through an icv injection 20 min before 0.5 nmol of the whole β-End 1–31 and morphine (10 nmols) were injected icv. The σ1R agonist PPCC, 3 nmol icv, was administered 10 min before β-End C-terminal peptides. * Indicates that at the corresponding time interval, the ligand being evaluated significantly altered β-End 1–31 or morphine analgesia. (**C**) A series of exogenous σ1R ligands (3 nmol) were icv-injected 20 min before 0.5 nmol β-End 1–31 was administered via icv in wild-type mice. The σ1R antagonist S1RA was also icv-injected before β-End 1–31 was administered icv in σ1R^−/−^ mice. (**D**) The effects of several doses of S1RA on morphine analgesia were studied in wild-type and σ1R^−/−^ mice. (**E**) A series of σ1R ligands were icv-injected 20 min before different opioids, and analgesia was evaluated 30 min after β-End 1–31 (0.5 nmol, icv), 15 min after DAMGO (0.1 nmol, icv), DPDPE (10 nmol, icv) and D-Ala^2^-deltorphin II (10 nmol, icv). (**A**–**E**) Bars are the mean ± SD of the pooled data from six mice. * Indicates that at the corresponding time interval, the ligand being evaluated significantly altered control analgesia (grey bars). ϕ Significant difference with respect to the group treated with the opioid and S1RA. ANOVA followed by the Holm–Sidak multiple comparisons test, *p* < 0.05, 1-β > 0.80.

**Figure 4 ijms-24-00582-f004:**
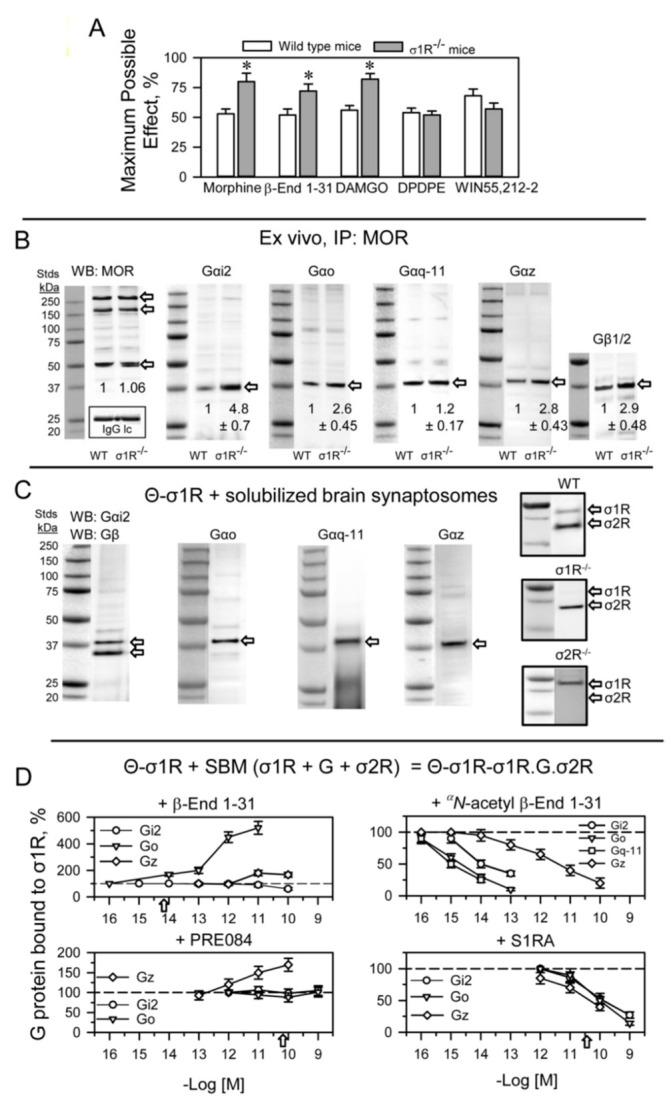
σ1R binds to G proteins and regulates MOR signaling in the mouse brain. (**A**) The effect of σ1R deletion on opioid- and cannabinoid-induced analgesia. Analgesic compounds were icv-injected into wild-type and σ1R^−/−^ mice, and analgesia was evaluated 30 min after morphine (3 nmol) and β-End 1–31 (0.5 nmol), 15 min after DAMGO (0.1 nmol) and DPDPE (10 nmol) and 10 min after WIN55,212-2 (15 nmol). The bars are the mean ± SD of eight mice per strain and treatment. * Significant difference with respect to the effect of the analgesic substance in the wild-type group. ANOVA followed by the Holm–Sidak multiple comparisons test, *p* < 0.05, 1-β > 0.80. (**B**) Coprecipitation of G proteins with MORs. Solubilized mouse brain cortical membranes enriched in synaptosomes (SBM) from CD1 wild-type (WT) and σ1R^−/−^ mice were solubilized with sonication in the presence of 1% NP-40. Non-soluble debris was eliminated, and the samples were then incubated overnight at 4 °C with affinity-purified biotinylated IgGs that were raised against extracellular regions of the MOR. Protein complexes that were immunoprecipitated with streptavidin agarose were extensively washed and then resolved by SDS-PAGE. The target G proteins, alpha and beta subunits, were visualized by Western blotting (for further details, see Section 4). IP: immunoprecipitated protein, WB: Western blot. Detection of MORs: these proteins were detected with an anti-MOR antibody directed to the C-terminal sequence, a different amino acid sequence than that used for immunoprecipitation (second extracellular loop-2EL). In parallel blots, an antibody directed to the light IgG chain of the anti-MOR antibodies used for immunoprecipitation of MORs provided a loading control for the samples in the gels. For each group, WT and σ1R^−/−^, the densities of the MOR-related bands indicated by the arrows, were pooled, and the computed value of the ko mice was referred to that of WT mice (assigned an arbitrary value of 1 value of 1). The immunosignals of Gα subunits coprecipitated with MORs were referred to those of the MORs. Data are the mean ± SD of three determinations. ANOVA followed by the Holm–Sidak multiple comparisons test, *p* < 0.05, 1-β > 0.80. (**C**) Pull-down assays using mouse SBM. Cerebral cortex fractions enriched in synaptosomes from CD1 WT mice were solubilized with RIPA buffer and incubated for 2 h with recombinant σ1Rs covalently attached to agarose-NHS (Θ-σ1R). Agarose pellets containing the bound proteins were resolved by SDS-PAGE, and the G protein α subunits were probed in Western blots. The capture of σ1Rs and σ2Rs by Θ-σ1R was also determined in SBM from WT, σ1R^−/−^ and σ2R^−/−^ mice. The assays were repeated twice, producing comparable results. Representative blots are shown. (**D**) Effect of ligands on the association of endogenous G proteins present in mouse SBM with Θ-σ1R oligomers produced in vitro (see Section 4). The Θ-σ1Rs were incubated with the SBM in the absence and presence of increasing concentrations of σ1R ligands. The G proteins that remained bound to Θ-σ1R oligomers were determined as in (**C**). The arrows indicate the affinity of the ligand for σ1Rs as determined through in vitro assays. For each compound and concentration studied data are the mean ± SD, n = 3. Data were analyzed by pairwise Holm–Sidak multiple comparison tests following ANOVA: *p* < 0.05, 1-β > 0.80.

**Figure 5 ijms-24-00582-f005:**
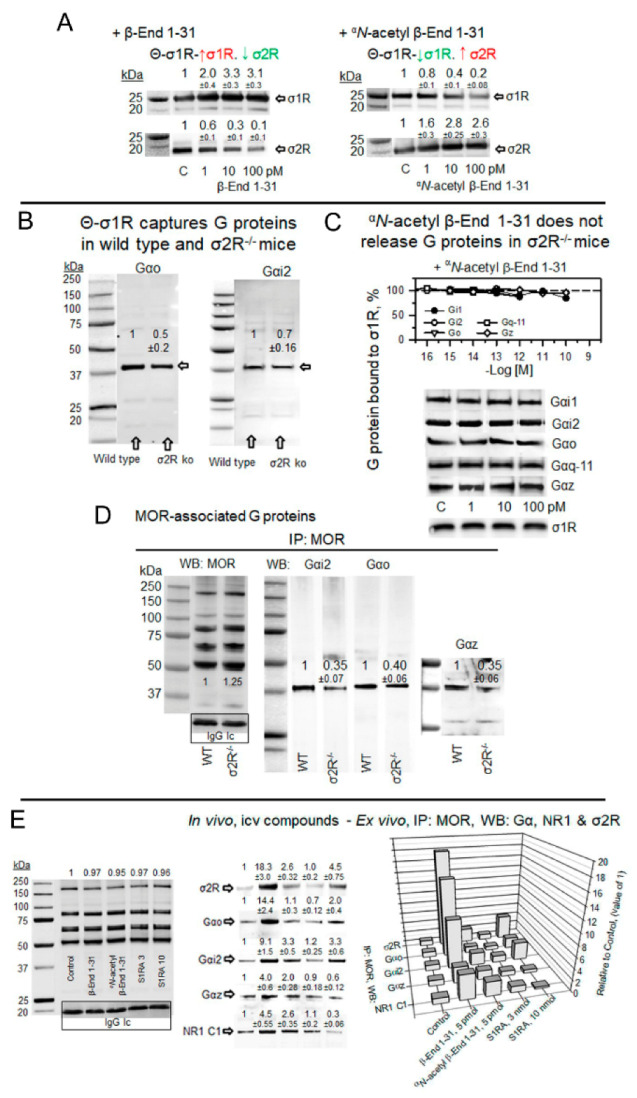
^α^*N*-acetyl β-End 1–31 promoted and β-End 1–31 reduced the exchange of G proteins for σ2Rs in σ1R oligomers. Implications in MOR signaling. (**A**) Effect of β-End 1–31 and ^α^N-acetyl β-End 1–31 on the association of σ1R and σ2R from mouse cerebral cortex with agarose-σ1R oligomers. Details are shown in Figure 4C,D. β-End 1–31 increased the presence of σ1R but diminished the σ2R association. In contrast, ^α^*N*-acetyl β-endorphin 1–31 dissociated the σ1R but promoted the σ2R association. The σ1R and σ2R immunosignals obtained in the absence of β-End 1–31 and ^α^N-acetyl β-End 1–31 were used as control (C, assigned an arbitrary value of 1) to which we referred those observed in the presence of increasing concentrations of the pituitary peptides. (**B**) Θ-σ1R captured G proteins in CD1 wild-type and σ2R^−/−^ SBM. Ex vivo fishing assays were performed as described in Figure 4C. The data from σ2R^−/−^ solubilized cortical tissue were referred to the wild-type control (arbitrary value of 1). (**C**) The σ1R oligomers captured G proteins from σ2R^−/−^ mouse SBM. For each Gα subunit and the σ1R, the immunosignals in the absence of the acetylated peptide were the controls (C, value of 1). In this scenario, ^α^*N*-acetyl β-End 1–31 did not produce significant alterations and thus failed to release G proteins and σ1Rs from σ1R oligomers. (**D**) Coprecipitation of G proteins with MORs in CD1 wild-type and σ2R^−/−^ mice. MORs were immunoprecipitated (IP) from mouse SBM. Coprecipitated proteins were detached from MORs and analyzed by Western blotting with antibodies directed to Gαi2, Gαo, Gαz and MOR proteins. (**E**) Effect of in vivo icv injection of β-End 1–31, ^α^*N*-acetyl β-End 1–31 and S1RA on the association of PAG MORs with G proteins, σ2Rs and NMDAR NR1 C1 subunits. β-End 1–31 (5 pmol), ^α^*N*-acetyl β-End 1–31 (5 pmol) and S1RA (3 nmol and 10 nmol) were icv-injected, and the mice were sacrificed 60 min post-treatment to obtain the PAG synaptosomal fraction. The MORs were immunoprecipitated from the solubilized membrane preparations, and the coprecipitated target proteins were analyzed by Western blotting. Further details are provided in the Section 4 and Figure 4B. Representative blots are shown. The immunosignal provided by the light chain of the affinity-purified anti MOR IgGs was used as SDS-PAGE loading control. No significant differences were observed between IP MORs corresponding to control and treatments. Thus, for each in vivo treatment, the immunosignals of the different Gα subunits and the NR1 C1 protein coprecipitated with MORs were referred to as the saline control group (assigned an arbitrary value of 1). (**A**,**B**,**D**,**E**) Data are the mean ± SD, n = 3. Data were analyzed by pairwise Holm–Sidak multiple comparison tests following ANOVA: *p* < 0.05, 1-β > 0.80.

**Figure 6 ijms-24-00582-f006:**
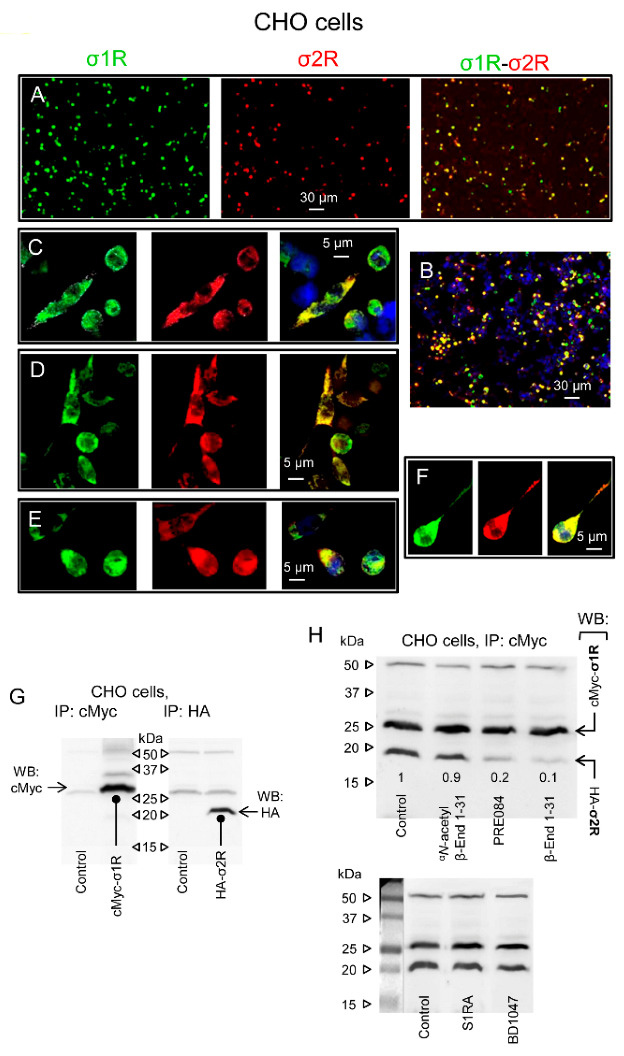
Physical interactions between σ1Rs and σ2Rs. Effect of β-End 1–31 and ^α^*N*-acetyl derivative. (**A**) Single-wavelength and merged images of CHO cells cotransfected with cMyc-σ1R and HA-σ2R. The cells were fixed 48 h after transfection and analyzed by confocal laser scanning microscopy. Transient expression of proteins within the transfected cells was detected with anti-cMyc (green) and anti-HA (red) antibodies (original magnification 10×). The fluorescent signals reflect the expression of the proteins of interest. Scale bar: 30 µm. (**B**) Nuclei stained with DAPI (blue). (**C**–**F**) Single-wavelength and merged images are enlargements of individual cells. Scale bar: 5 µm. Lower panels: (**G**) Immunoprecipitation (IP) of tagged sigma receptor types from cotransfected CHO cells. Biotin-conjugated anti-cMyc and anti-HA antibodies precipitated sigma receptors as detected by Western blot analysis using antibodies that recognize either tag of the sigma receptor proteins. (**H**) σ1R/σ2R coprecipitates from cultures incubated with β-End 1–31 (1 nM), ^α^*N*-acetyl β-End 1–31 (1 nM), the σ1R agonist PRE084 (100 nM) and the σ1R antagonists S1RA (100 nM) and BD1047 (100 nM). The control was assigned an arbitrary value of 1, and the changes observed in the test groups were referred to as the control. Representative blots are shown.

**Figure 7 ijms-24-00582-f007:**
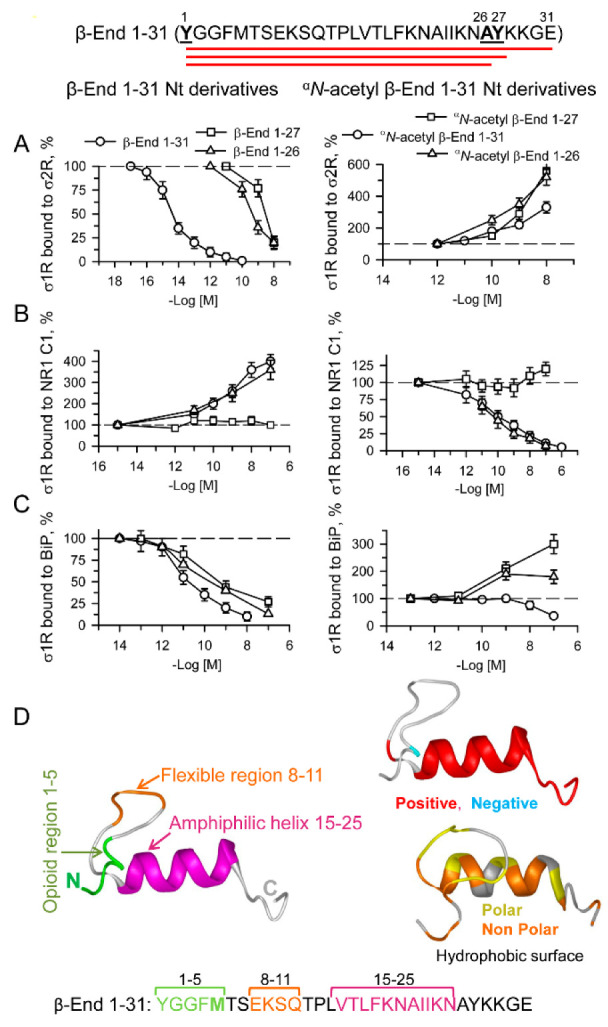
Effect of β-End 1–31, ^α^*N*-acetyl β-End 1–31 and their N-terminal derivatives on σ1R-σ2R, σ1R-NR1 C1 and σ1R-BiP associations. Amino acid sequence of β-End 1–31 showing derivatives 1–26 and 1–27. (**A**) HaloLink-attached σ2Rs were incubated for 30 min at RT in the presence of σ1Rs (100 nM) in 300 μL of 50 mM Tris-HCl (pH 7.4), 0.2% CHAPS, and 2.5 mM CaCl_2_. After removing the unbound σ1R proteins, the effects of increasing the peptide concentrations on σ1R-σ2R associations were studied. At the end of the incubation, protein complexes bound to HaloLink-σ2R were obtained by centrifugation, washed three times, solubilized in 2× Laemmli buffer containing β-mercaptoethanol, resolved by SDS-PAGE, and analyzed by Western blotting (for more details, see Section 4). This protocol was also used to assess the effect of ligands on (**B**) σ1R-NR1 C1 and (**C**) σ1R-BiP associations. Data were compared to that of the control, which was obtained in the absence of ligands. (**A**–**C**) Data are the mean ± SD, n = 3. (**D**) The β-End 1–31 structural models were predicted by Novafold (DNASTAR Lasergene protein v.17, Inc., Madison, WI, USA). Ribbon model: The 3D structure, charge distribution and hydrophobicity map of β-End 1–31 are shown. Positive charges are shown in red and negative charges in blue. Polar surfaces are shown in yellow, and nonpolar surfaces are shown in orange. Linear model: the opioid (1–5) and flexible region (8–11) sequences are shown in green and orange, respectively, and the amphiphilic helix (15–25) is shown in pink.

**Figure 8 ijms-24-00582-f008:**
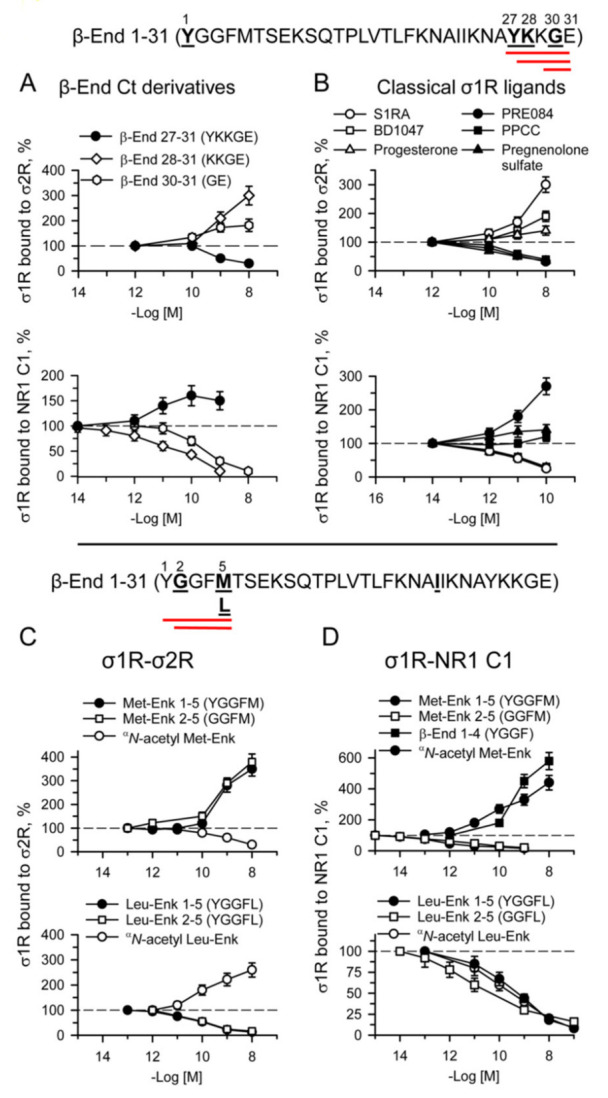
Effect of β-End 1–31 C-terminal derivatives, enkephalins and classical σ1R ligands on σ1R-σ2R and σ1R-NR1 C1 associations. Upper panel: amino acid sequences of physiological β-End C-terminal derivatives. The effects of increasing concentrations of (**A**) β-End C-terminal 27–31, 28–31, 30–31, and (**B**) classical σ1R ligands, such as the antagonists S1RA, BD1047, and progesterone, and the agonists PRE084, PPCC, and pregnenolone sulfate, were studied on σ1R-σ2R and σ1R-NR1 C1 associations. Data were compared to that of the control, which was obtained in the absence of ligands. Lower panel: amino acid sequences of β-End N-terminal derivative 1–5 [methionine-enkephalin (Met-Enk)] and its aminopeptidase product 2–5. The sequences of 1–5 leucine-enkephalin (Leu-Enk) and 2–5 metabolite are also indicated. (**C**,**D**) Effects of Met-Enk, Leu-Enk, their chimeric ^α^*N*-acetylated forms and 2–5 metabolites on σ1R-σ2R and σ1R-NR1 C1 associations. Data were compared to that of the control, which was obtained in absence of ligands. (**A**–**D**) For details, see Section 4 and Figure 7. Data are the mean ± SD, n = 3.

## Data Availability

The data and materials of the manuscript are available upon reasonable request.

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
