# Peer review of "αN-Acetyl β-Endorphin Is an Endogenous Ligand of σ1Rs That Regulates Mu-Opioid Receptor Signaling by Exchanging G Proteins for σ2Rs in σ1R Oligomers"

_ijms, 2022, doi:10.3390/ijms24010582_

Round 1

Reviewer 1 Report

This study applies a number of experimental procedures to test the mechanisms by which N-acetyl-endorphin reduces vascular infarct, convulsive syndrome, allodynia caused by various insults, opposite to the effects of endorphin. Also, N-acetyl-endorphin blocks the3 antinociceptive effects of several opioid agonists but not DAMGO. The experimental results demonstrate that the sigma1R protein is involved, mediating sequestration of G proteins and sigma1R oligomerization. The authors conclude that the previously identified “pharmacological β-endorphin specific epsilon receptor is a σ1R-regulated MOR and that β-Endorphin and αN-acetyl β-Endorphin are endogenous ligands of σ1R.”

These results are interesting and well supported by the experimental results. The authors build on their 1992 paper showing that acetyl-endorphin alleviates opioid withdrawal symptoms, and a aper by Cavun et al (2005) on the effects of endorphin30-31. These results have the potential to clarify some of the perplexing effects of opioid and related ligands.

The main issue with the manuscript is the lengthy presentation of the results and conclusions. For example, the Introduction appears to summarize the results again, and without references, making it difficult to discern the innovation inherent to the new data. Similarly, the Discussion rambles on, so that it requires careful reading to dissect what results and concepts are novel. A much more focused Discussion would enhance the impact of the paper.

Author Response

This study applies a number of experimental procedures to test the mechanisms by which N-acetyl-endorphin reduces vascular infarct, convulsive syndrome, allodynia caused by various insults, opposite to the effects of endorphin. Also, N-acetyl-endorphin blocks the3 antinociceptive effects of several opioid agonists but not DAMGO. The experimental results demonstrate that the sigma1R protein is involved, mediating sequestration of G proteins and sigma1R oligomerization. The authors conclude that the previously identified “pharmacological β-endorphin specific epsilon receptor is a σ1R-regulated MOR and that β-Endorphin and αN-acetyl β-Endorphin are endogenous ligands of σ1R.”

These results are interesting and well supported by the experimental results. The authors build on their 1992 paper showing that acetyl-endorphin alleviates opioid withdrawal symptoms, and a aper by Cavun et al (2005) on the effects of endorphin30-31. These results have the potential to clarify some of the perplexing effects of opioid and related ligands.

The main issue with the manuscript is the lengthy presentation of the results and conclusions. For example, the Introduction appears to summarize the results again, and without references, making it difficult to discern the innovation inherent to the new data. Similarly, the Discussion rambles on, so that it requires careful reading to dissect what results and concepts are novel. A much more focused Discussion would enhance the impact of the paper.

We thank you for your kindness and appreciation of our effort in preparing the manuscript. The constructive comments made by the Reviewer were undoubtedly very stimulating. We acknowledge your expert advice, and we thank you for your demonstrated interest and for dedicating your valuable time to reviewing our article. 

We have removed the summary of results from the Introduction and have attempted to improve the readability of the Discussion. The proposal of beta endorphin as an endogenous σ1R ligand requires the use of different experimental approaches which inevitably generates a considerable amount of data. Given the novelty of the finding, the discussion of the results necessarily requires a certain amount of text.

We thank you again for your interest and would be very pleased if you would recommend our work for publication in IJMS.

Reviewer 2 Report

This reviewer would like to congratulate the authors on making major progress in understanding the biological relevance of N-acetyl beta endorphin, and for their identification of several therapeutic applications of this endogenous peptide. Furthermore, the mechanistic biochemical studies are an elegant approach towards verifying the involvement of sigma containing protein complexes in the action of this peptide. The manuscript has some weaknesses, some of which may be readily addressed if data was previously obtained but omitted during writing. Some language also should be qualified to avoid overstating conclusions beyond available data. Please see the attached document for itemized comments.

Author Response

This reviewer would like to congratulate the authors on making major progress in understanding the biological relevance of N-acetyl beta endorphin, and for their identification of several therapeutic applications of this endogenous peptide. Furthermore, the mechanistic biochemical studies are an elegant approach towards verifying the involvement of sigma containing protein complexes in the action of this peptide. The manuscript has some weaknesses, some of which may be readily addressed if data was previously obtained but omitted during writing. Some language also should be qualified to avoid overstating conclusions beyond available data. Please see the attached document for itemized comments.

We would like to express our gratitude for your time and appreciation for our effort in preparing this manuscript. PSB and JGN, attended the initial studies on β-endorphin by joining the E. Way-H.H. Loh (UCSF-MC) group in the early 1980s. Our studies continued in the laboratory of A. Herz (Max-Planck, Munich) where the epsilon receptor was proposed. After our return to Spain (Instituto Cajal, CSIC, Madrid) we started this study of several years that we now believe can be disseminated. Certainly we have more data that for reasons of space or to improve the clarity of the graphs have not been incorporated in the manuscript. The English language of the current version of our article has been reviewed by Springer Nature Editing Service. However, we can request the offered by the Editorial to accommodate the IJMS standards.

Introduction:

Overall, the introduction is well written and provides valuable background. Here are 3 suggestions for improvement.

1- The sentence at line 71, might better reworded thus: “Thus, β-End 1-31 strongly stimulates GTPγS binding in wild-type membranes but show no detectable effects in membranes from triple MOR, DOR, and KOR knockout mice.” “However, loss of β-End 1-31 stimulated GTPγS binding in membranes from triple MOR, DOR, and KOR knockout mice suggested the epsilon receptor may be comprised of one or more of the proteins encoded by the genes for MOR, DOR, and/or KOR.” The first sentence of the paragraph that follows might be adjusted after this clarification. The word “thus” in the sentence at line 71 suggests the triple KO result follows from the previous sentence, but in fact I suspect it did not really follow, and that result was probably somewhat surprising (and maybe even disappointing) for epsilon researchers at that time.

Thank you, the sentence has been changed.

2. At line 85, the statement “the ε receptor was defined as a benzomorphan binding site that was distinct from MOR, DOR, or KOR” could be qualified to “the ε receptor was defined as a benzomorphan binding site that was distinct from canonical MOR, DOR or KOR” to be more compatible with the citation of loss of epsilon site activity in triple KO data.

Changed

3. The rationale for analyzing only Gαi2 and omitting Gαi1 and Gαi3 is not clearly stated. In fact, there is a study in which Gαi2 knockout mice demonstrated no change to morphine analgesic sensitivity (Lambert et al 2011 Neuropharmacology 21654736), Provision of a reference for how this rationale was developed would be helpful. The rationale might also be mentioned in results section 2.5.

From the four subfamilies of Gα subunits, αs αi αq α12 (Downes & Gautam, Genomics, 62:544-552, 1999, PMID: 10644457), we analyzed representative Gα subunits of these regulated by MORs in the production of opioid analgesia when injected by icv route. Therefore, from αi we include pertussis toxin sensitive Gαi2 and Gαo, and the pertussis toxin-insensitive Gαz; from αq the pertussis toxin insensitive Gαq was selected. Including more classes of G proteins is beyond the scope of this manuscript. It would be interesting to explore the other Gi proteins, Go1/Go2 subclasses, G12, and Gs. 

Since the final part of the Introduction has been reduced, the rationale of Gα study is now mentioned in Results.

Lambert’s paper deals mainly with Gαo ko mice and also includes data from Gαi2 and Gαi3 ko mice. To compare data from different studies it is necessary that both be performed using similar protocols and the Lambert study differs clearly of ours in which the temporary reduction of Gα subunits is carried out by injection of antibodies, or oligodeoxynucleotides. In addition, opioids were also studied via icv to determine supraspinal analgesia. We usually study the time course of the analgesic effects of opioids and also their dose-dependent effects determined when analgesia peak. Lambert’s study uses peripheral administration, i.p. and cumulative doses of opioids and this approach should address pharmacokinetic and pharmacodynamic limitations. On the other hand, the elimination of such abundant and functionally relevant Gα proteins is usually compensated by others that exhibit nearby functions.

Lambert’s study found significant decreases in the expression of certain γ and β subunits. These changes must necessarily alter the relationship of G proteins remaining with GPCRs. By reducing the expression of Gα subunits, the signaling capacity of their coupled βγ dimers can be altered. Therefore, reductions in Gα could also alter analgesia through changes in the function of βγ dimers, and no one has done much work on this subject. Several papers from different groups have shown that Gi2 proteins are critical for “supraspinal” morphine analgesia (icv route). This has been reported with oligos in mice by Raffa’s group (PMID: 7925586), and in rat by G.W. Pasternak’s group (PMID: 8592651). The latter paper demonstrated that Gi2 is critical for morphine analgesia but not for that of its morphine-6 beta-glucuronide (M6G) metabolite, and the contrary for Go. It therefore possible that in Gα ko mice the ratios of MOR subtypes changes and thus the relevance of other G proteins in opioid analgesia. See also ours (PMID: 10565001).

I am sure you are aware that not only our studies but also those of other groups based on the temporary reduction of Gα proteins have revealed the role of Gi2 proteins in MOR analgesia. To avoid confusing the subject matter, we try to make sure that a ko mouse can provide valuable information. So before using ko mice in complex studies we proceed to the temporary reduction of the target protein in the wild type and the comparison of both phenotypes, knockdown and knockout, see e.g. “The Sigma 2 receptor promotes and the Sigma 1 receptor inhibits mu-opioid receptor-mediated antinociception” Mol Brain, 13: 150, 2020, PMID: 33176836.

4. This early study may be cited (Akil et al Sciencec 1985 PMID 3155575), to put into context how long it has been known that N-acetyl beta-endorphin is released in vivo, despite lack of progress on the understanding of what its activities substrates are, as have been elucidated here.

We thank you for your indication; it is relevant to the subject and is now cited

The results section provides data supporting novel findings which are of high interest, but some aspects of the writing need further clarification or adjustment, and provision of additional data would bolster conclusions asserted by the authors.

Results:

5. How were the interventions (peptides, ligands) delivered in the MCAO study shown in Fig1A? ICV? Please clarify the route and method of administration.

All compounds used in the pMCAO test Fig. 1A were administered by icv route as indicated in methods (line 880, first version). This information is now included in the legend to the figure. We do not include the detailed procedure because this is considered plagiarism: “To facilitate selective and straightforward access to their targets, the compounds were injected (4 µL) into the lateral ventricles of mice as described previously [Haley & McCormick, Br J Pharmacol Chemoter, 12, 12, 1957, PMID: 13413144; Rodríguez-Muñoz et al., Antioxid Redox Signal, 22, 799-818, 2015 PMID: 25557043). Injections were performed with a 10 µL Hamilton syringe at a depth of 3 mm at a point of 2 mm lateral and 2 mm caudal to the bregma. The compounds were infused at a rate of 1 µL every 5 s, after which the needle was maintained in place for an additional 10 s and we only refer to our previous studies when it was initially published.

6. In figure 1B, there is a statistically significant improvement in the NMDA seizure model, with N-acetyl beta-endorphin treatment in wt for hypermobility and rearing, but not clonic convulsions and not tonic seizures. The text of the results (~line 139) claims improvements ssfor all four phenotypes, but the data shown does not support for the latter two, only the first two. The text of the results needs to be corrected. Similarly, the statement that PPCC blocked the N- acetyl-beta endo alleviation of the convulsive syndrome (line 147) should be clarified to state that PPCC blocked two of the features of NMDA reponse that were alleviated by N-acetyl-beta endo but not the other features suggesting distinct neural substrates underlie distinct components of the seizure model.

Following your advice, these lines were changed “αN-acetyl β-End 1-31 partially protected the mice from hypermobility and convulsive rearing, while only a tendency to diminish clonic convulsions and tonic seizures was observed, suggesting distinct neural substrates underlie distinct components of the seizure model. PPCC prevented αN-acetyl β-End 1-31 from alleviating these sings of NMDA-induced convulsive syndrome”

7. In section 2.5, it is mentioned that G protein association is increased and indeed the figure 4B suggests this may be true, but this point is not determined rigorously. Quantitation and statistical testing from n>=3 animals for each genotype should be provided to make this claim (~line 283).

Thanks for your comment, we provide SDs in the graph and the “n” is described in the legend to the figure.

8. There are no loading controls for the western blots presented. Actin or some other commonly accepted loading control should be used when the intent is to use the western blot method and band intensities to make assertions about differences in the bands between treatments/genotypes. Similarly, loading controls for input materials put into the IP studies would support that the 2 conditions under comparison were similar at the start of the IP process. This issue is present in data in figure 4 through 8.

The controls and details of our Western blots studies have been described several times in previous publications. This time we believed that such information may interfere with the good visualization of the data represented in the figures. Following your instructions we have incorporated/described the controls you request.

In the ex vivo IP studies we do not consider reliable controls those proteins that may eventually accompany the target protein since their presence is modified by many factors such as in vivo treatments or during the immunoprecipitation process of the samples. As they are immunoprecipitations, the reliability of the load is mainly the low dispersion of replicates. However, in the comparison between two experimental situations the signal of the heavy or light chain of biotinylated IgGs used in immunoprecipitation is useful.

In Fig. 4B, the loading control of MOR IP is the reproducibility of the triplicate (variation typically less than 10%) and in the comparison between different treatments or experimental situations, we use the signals of the IgG’s light chain (lc) used in the immunoprecipitation (see e.g., Mol Pain 3: 19, 2007, PMID: 17634133). These are now shown in the left panel of Figure 4B. MOR signals are now the control to which we refer the Gα signals that accompany the receptor during IP.

Fig. 4C provides qualitative information on the capture of a series of proteins by the σ1R coupled to agarose, as well as the presence or not of one type of σ receptor.

Fishing assays (ex vivo): the loading control corresponding to an experimental situation is essentially the variability of the replicates that in our criterion should not exceed 10%; this control is applicable to Fig. 5B. In trials with increasing concentrations of ligands in which dose/concentration-changes are observed it is evident that the probability that these variations are random/non-specific and caused by uneven loading is certainly low. Similar situation in Figs. 4D, 5A and 5C.

Fig. 5D, the light chain of IgGs used in immunoprecipitation is the loading control for MOR IP. Now, the receptor signals are controls of co-immunoprecipitated Gα subunits.

Fig. 5E, as in 5D, the MOR IP control is the light chain of the IgGs used in immunoprecipitation, and those of the receptor for each treatment serve as a control to which the signals of the co-precipitated proteins refer (see legend to the Figure 5)..

In vitro interaction assays: Figs. 7 & 8, the fishing protein would be a nice loading control; however it is covalently attached to the agarose and remains at the stacking part of the gel. Thus, as mentioned above for a given experimental situation, control or different concentrations of the σ1R ligand, the low variation of the replicates are an index of correct loading of the sample, and in the comparison between samples, dose-dependent changes reduce the possibility that the observed variations be unspecific. In these figures, data are presented as mean±SD, n=3. To improve visualization of the different but very close curves (Figure 4D, Figures 7 & 8) we have omitted the symbols indicating significant differences vs the control (absence of compound in the assay medium). 

9. Section 2.5, the IP with anti-MOR antibody may be contaminated by pull down of other proteins, as the antibodies for MOR-1 are well known to have specificity problems (such as described in Niwa BJA 2012 PMID 22337965). It would be helpful to know the pulldown technique is relatively specific for MOR-1, by including results obtained from an MOR-1  KO mouse brain.

This reference is not related to our experimental approach. The authors study the direct detection of MOR/KOR after forcing their expression in CHO cells. The battery of antibodies they use includes raw sera, purified antibodies (without specification of the method used), etc. In nervous tissue, the low level of expression of opioid receptors compromises their direct immunodetection in transfer membranes, in which MOR binding to the corresponding restricted area can be prevented by competition with other proteins of similar electrophoretic mobility but much more abundant in synaptosomes. As a result, most commercial antibodies do not detect MOR protein in brain tissue.

To circumvent this drawback, detection of neural MORs requires of their enrichment through immunoprecipitation from solubilized brain synaptosomes. The quality and specificity of our MOR IP procedure has been addressed in previous reports. e.g., JBC 280: 8951-8960, 2005, PMID: 15632124. Following this procedure, our antibodies directed to different regions of the MOR sequence, N-terminal (Nt) and second external loop (2EL) provided coincident immunosignals (Neuropharmacology 48:853-868, 2005, PMID: 15829256; Neuropsychopharmacology 30:1632-1648, 2005, PMID: 15827571). These signals are not obtained in NG10812 cells not expressing this receptor (Mol Pharmacol 47: 738-744, 1995 PMID: 7723734), and MOR ko neural tissue (Neuropharmacology 48:853-868, 2005, PMID: 15829256; Neuropsychopharmacology 30:1632-1648, 2005, PMID: 15827571).

The IP of transmembrane proteins such as GPCRs requires targeting of extracellular domains free of cytosolic interactions with other proteins. However, post-translational modifications of extracellular domains such as glycosylation may complicate MOR immunodetection. Thus, in size-directed chromatography several weight forms of the MOR are detected due to glycosylation (Mol Pharmacol 47: 738-744, 1995 PMID: 7723734). Because the N-terminal MOR sequence includes a potential glycosylation site that may prevent the binding of IgGs to this region, it is advisable the use of the 2EL MOR antibody.

Thus, in methods (4.6) we refer to our previous characterization of the MOR antibody and the reliability of the immunoprecipitation procedure used.

10. Are the data for Figure 4D performed with triplicates? If so, please add error bars to the plot and statistical tests. Otherwise, the determinations might be due to chance, and the inferences qualified.

In the revised manuscript this information is incorporated in the figure and legend.

11. The data in figure 5 also are all lacking triplicates and error bars and statistical analyses, precluding strong conclusions about the trends inferred from these studies.

We are providing this information in the Figure and legend.

12. Section 2.5, line 356, it is claimed that the overexpression of tagged sigma1 and sigma2 are localized to plasma membrane. This assertion should be supported by confocal microsopy showing staining at the cell edges, ideally with a plasma membrane marker in another channel. Without such data, it is not credible to know whether the label is in Golgi, or even perhaps cytoplasm. Publications report sigma protein as an ER localized protein (for example, Sharma JBC 2021 PMID 34648767, though other reports abound).

This is a critical point and we would like to comment on it. The work you mention and also other works try to extrapolate the findings in cell lines to the function of neuronal cells in adult tissue. Although elegant and technically flawless, experience teaches us that the physiology and abnormalities of the neuronal network as neurodegenerative processes are hardly reproducible in studies in non-neuronal cell lines, especially when the expression of proteins such as the σ1R and mutated forms are forced.

At the Cajal Institute, which is dedicated to the study of the nervous system, the groups that study plasticity, development and neurodegenerative diseases, none of them base their activity on cell lines. Instead, they use primary cultures of neurons and glial cells, as well as cultures of adult neurons. The complexity of adult neuronal structure completely escapes what can be modeled in non-neuronal cell lines

Answering your question, by means of confocal microscopy and triple labelling we observed the presence of σ1Rs in the plasma membrane of brain PAG neurons. In the neural surface this receptor co-localizes with MOR and glutamate NMDA receptors (The σ1 receptor engages the redox-regulated HINT1 protein to bring opioid analgesia under NMDA receptor negative control, Antiox Redox Signal 22: 799-818, 2015, PMID: 25557043). Importantly, this study showed that icv administration of recombinant σ1Rs restored MOR-NMDAR functional coupling in σ1R ko mice, thus suggesting their insertion from ventricular space into the neural plasma membrane.

We do not question the presence of σ1Rs in ER, but at least in neural cells it is also found in the plasma membrane or associated structures where it plays an important signaling role and is accessible from the extracellular space to various ligands, some as β-endorphin 1-31 which could be one physiological ligand. Thus, simpler models such as cell lines must help explain the neural physiology, instead of the neural physiology having to accommodate what is observed in these models.

If we are correct, there is no report demonstrating that extracellular ligands, with the exception of neurosteroids, penetrate neural cells to regulate the σ1R at the ER or at the mitochondrial interface. Without such proof, it remains possible that ligands bind to this receptor on the cell surface or in associated structures on the inner face of the plasma membrane and from there the activated σ1R could translocate to other cellular compartments or simply internalize its activation through specific signaling pathways.

To properly manage the therapeutic potential of sigma receptor ligands, it is necessary to demonstrate how these ligands reach their receptors in vivo. If they are internalized, is this passive process (not in the case of peptides such as endorphins) or do they bind to sigma receptors or another carrier? Without this knowledge their dosing can be erratic and difficult to control their pharmacodynamics/pharmacokinetics.

 Years ago, we observed the therapeutic potential of icv recombinant  Gα subunits, and we had to demonstrate that myr-Gα subunits coupled to colloidal gold particles when icv-injected reached the cytosolic compartment of neural cells. These proteins were detected by electron microscopy (Myr-Gi2α and Goα subunits restore the efficacy of opioids, clonidine and neurotensin giving rise to antinociception in G-protein knock-down mice, Neuropharmacology 38: 1861-1873, 1999, PMID: 10608281).

Therefore, without questioning that after their forced expression in CHO cells the σ1R and σ2R are located in cytosol/ER, I believe that they are also found in the plasma membrane where extracellular σ1R antagonists prevent or reverse the association between both receptors.

A few lines are included at the end of the revised 2.5. section.

13. Section 2.5, line 362, it is stated that the overexpression of tagged constructs are comparable between the sigma1 and sigma2 constructs. However, the reagents used to detect them should not be used to make quantitative inferences between different proteins with different tags detected with different detection reagents.

Thank you for your pertinent remark. This phrase has been withdrawn.

14. In section 2.7, are the data presented in figure 7A through 7C averaged triplicates? Also, are there loading controls available for evaluation for the inputs to the pulldown assays? If such controls are not available, it is difficult to rigorously assert the quantitative effects which are suggested.

The fishing protein σ1R is covalently coupled to the agarose and the prey (σ2R, NR1 C1 or BiP) is in solution along with the β-endorphin derivative. The samples are loaded after the wash-resuspension cycles to remove the free prey, and contain agarose-σ1R.prey The Laemmli SDS buffer simply removes the prey coupled to the s1R-agarose that during SDS-PAGE remains on top of the gel.

Therefore, σ1R cannot be used as a load control. Data obtained through such tests with recombinant proteins are accepted and therefore used by many other laboratories as well. The low dispersion of the replicates that is typical of in vitro protein interaction assays makes them useful when answering simple questions like the ones we are asking here. Of course, small changes may be of limited confidence, but those that increase dose-dependently with interfering ligands are relevant. The data of n=3 ±SD are now presented.

15. The experiments of figure 7 and 8 use recombinantly expressed purified sigma1 and sigma2, but the purity of the extracts are not shown. Please provide Coomassie or similar non-specific stains of the reagents employed for these studies as supplemental data, and representative approximate purity of the overexpressed and purified proteins that are used in the experiments, that the reader can appreciate the purity of the reagents from which numerous conclusions are based. Also, have control experiments been performed to ensure that the tags that are used in the experiment are not interacting with the peptides?

This information is shown in the revised manuscript, Figure S11.

16. At line 499, there is a typo, peptapeptide should pentapeptide.

Corrected

17. There is a line still extant in this version that appears to be a placeholder left by authors during drafting (lines 525-527).

Corrected

Overall, the discussion is rather long and might benefit from editing for length.

17. In the discussion, the authors claim that Gz protein is essential for morphine analgesia (line 840). For this statement, they self-cite their antisense knockdown studies demonstrated the importance of Gαz, but omit an important definitive study by Leck et al Neuropharmacology 2004 PMID 15033343 that demonstrated that acute morphine analgesia is intact in KO mice, albeit attenuated in Gαz ko female mice. The two strategies have their weaknesses, with possibility of off target effects with antisense, and compensation in constitutive KO mice. Therefore, whether Gz is essential for acute morphine analgesia warrants more rigorous study with modern conditional ablation techniques. Nevertheless, the Leck study should be incorporated into the discussion and the wording qualified.

We respectfully disagree with your comment that Leck’s paper is "important definitive" We are always willing to reconsider our knowledge and base it on probability and not on what at any given time might seem definitive. The Leck’s work is not at all comparable with ours, more so are others that are cited in that paper and that also describe loss of analgesia in Gαz ko mice (PMID: 10954748). The interpretation of Yang et al., of their observations is shared by us “Deletion of Gzα caused a limited shift in the dose-response curve to morphine; this suggests that mu-opioid receptors either may interact with other G proteins as well or Gz is linked to only a subset of the receptors that are involved in this response.”

I agree that the use of modern gene ablation approaches may shed light on some controversial issues. However, in pharmacological studies this must be accompanied by adequate protocols and well defined objectives. While we always use the icv route to reduce Gα expression and to inject the opioids, Leck and Yang studies use the systemic (intraperitoneal route) to administer morphine. Opioids passing the BBB given by i.p. or subcutaneous route reach spinal cord level much better than brain structures (PMID: 6686142 & 6092607). Since the Gz protein and its RGS regulators, RGSZ1 and RGSZ2 are far more abundant in supraspinal structures than in spinal cord (Neuropsychopharmacology, 30:1632-1648, 2005, PMID: 15827571), the MORs at spinal/ganglia level may better couple to other G proteins (as we suggest for MORs in RVD), probably pertussis-toxin insensitive such as Go. It would be nice whether these authors checked the sensitivity of their morphine analgesia to pertussis toxin.

Notwithstanding, Leck not only detects a reduction in morphine analgesia in Gαz ko females but also mentions a clear tendency to be significant in Gαz ko males. This probably implies that part of i.p. morphine analgesia was mediated by brain structures and that MORs couple and activate various classes of G proteins to promote antinociception. The use of cumulative doses of the opioid to perform dose-response studies, as these authors did, may alter MOR association with G proteins. This approach may be used in isolated organ studies where opioids are incorporated in a large volume of the Ringer’s buffer and thus pharmacokinetic has little influence, but in vivo development of tolerance and sequestering of the already injected opioid may shift the coupling of the GPCR to other classes of G proteins.

On the other hand, when the Gαz ko mice are rendered tolerant to morphine, Leck’s work and ours described the relevant role of Gz in this adaptive process. Leck’s work shows that the loss of the Gz leads to a greater tolerance to morphine, and ours (Cell Signal, 21:1444-1454, 2009, PMID: 19446022; PlosONE, 5, e11278, PMID: 20585660) demonstrated the interplay between Gi2 and Gz proteins with the first promoting tolerance to morphine and the second acting as a brake.

We are confident that our study on morphine analgesia and Gz proteins provides reliable information. We described that MORs couple to PTX-sensitive and insensitive G proteins. Supraspinal (icv) analgesia promoted by morphine, etorphine, β-casomorphin (1-4) and β-endorphin 1-31 have an important PTX-resistant component, while for that of DAMGO, FK33824, DAME and DADLE was much less evident. In this scenario, the second group of opioids reduced the analgesia of morphine (Eur J Pharmacol, 152, 357-361, 1989, PMID: 3220110). Similar results were ontained with N-ethylmaleimide (Eur J Pharmacol, 166, 193-200, 1989, PMID: 2676563). Using a nonisotopic, immunoelectrophoretic technique we showed that Gz was activated via MOR by morphine and β-endorphin 1-31 and poorly by DAMGO. On the other hand Gi2 was better activated by DAMGO than morphine (JPET, 281, 549-557, 1997, PMID: 9103543). The in vivo icv administration of IgGs directed to different classes of Gα subunits (Brain Res Mol Brain Res, 65, 151-166, 1999, PMID: 10064886) also linked morphine with mainly Gz and DAMGO with Gi2 proteins. Temporal reduction of one of these G proteins brings about antagonist effects between these MOR opioid ligands (Life Sci, 53, 381-386, 1993, PMID: 7902522; Life Sci, 55, 205-212, 1994, PMID: 7915394). The use of icv oligodeoxynucleotides against different Gα subunits also produced comparable results to these afore described (JPET, 291, 12-18, 1999, PMID: 10490881; Methods Enzymol, 314, 3-20, 2000, PMID: 10565001; Brain Res Bull, 54, 229-235, 2001, PMID: 11275413). Moreover, we demonstrated that opioid show biased activity after binding with diverse affinity to MORs coupled to distinct G proteins (Eur J Neurosci, 10, 2557-2564, 1998, PMID: 9767386). Notably, after temporal reduction of a class of G proteins, the MOR can regulate other classes of these transducer proteins and recover at least part of its analgesic activity (Neuropharmacology, 38, 1861-1873, 1999, 10608281). 

Of course, our present work is not the place to review and discuss in detail the issue of MOR regulation of G protein classes in analgesia. In the revised manuscript, we have introduced some lines in the Discussion in terms of the relevance of Gz in morphine supraspinal but probably not in spinal analgesia.

18. In the title and also the last paragraph of the discussion, it is claimed tσ2Rhat the beta-end 1-31 and N-acetyl-beta-end 1-31 bind to sigma1R. The data in Figure 7 support that these peptides interact with either sigma1R or sigma2R, to interfere with their interactions with each other, but there seems to be no data supporting the assertion that the binding is directly with sigma1 rather than with sigma2. The interference of interaction between sigma1R and NR1 C1, and with BiP is suggestive, but could be due to these peptides also binding to the sigma1 partners, rather than with sigma1 itself.

The demonstration with a binding assay between radiolabeled N-acetyl-beta-end 1-31, and sigma1 purified protein, or with wt vs sigma1 KO mice brain tissue, would be stronger support for binding of this peptide with sigma1R. Alternatively, a CETSA assay experiment, may be performed with purified sigma1 alone versus sigma2 alone, with mutagenesis of predicted interacting residues may also provide further support for the assertion of direct binding of this peptide to sigma1.

Our studies have addressed this issue from different perspectives, and we consider that the data obtained are compatible with the binding of β-endorphin 1-31 and N-acetylated derivative to the σ1R.

-The in vivo effects of these peptides are absent in σ1R ko mice.

-They alter the associations of σ1Rs with partner proteins such as σ2R, NR1 C1 and BiP.

-The above mentioned effects are shared by typical ligands of σ1Rs.

-β-endorphin 1-31, αN-acetyl β-endorphin 1-31 and physiological derivatives compete with typical σ1R ligands for their effect in vivo and in vitro models.

-The regulatory effects of αN-acetyl β-endorphin 1-31 on MOR analgesia are achieved at low fmols, and the disrupting effects of β-endorphin 1-31 on σ1R-σ2R and σ1R-BiP associations were observed at low fM and pM concentrations respectively. Similarly, αN-acetyl β-endorphin 1-31 disrupted σ1R-NR1 C1 complexes at pM concentrations.

-As you suggested we explored their effects on the associations of the ALS-related σ1R human mutant E102Q with other proteins (NR1 C1 & σ2R), and both, typical ligands and β-endorphin 1-31/ αN-acetyl β-endorphin showed poor o no capacity to alter these complexes. See 2.7 and new Figures S5 & S6.

Radioligands assays are no longer allowed at our Institute, and we are encouraged to use alternatives, but no one provides them. On the other hand, Y27 β-endorphin iodine 1-31 and the radiolabeled derivative N-acetyl are currently not available. Binding studies with these neuropeptides would be very interesting. However, we cannot rule out that in synaptosomal preparations in addition to σ1R they also bind to other proteins such as Gα. Therefore, σ1R gene ablation will not provide definitive binding data and may be necessary computational analysis whose conclusions are always subject to interpretation.

Trials such as CEPSA or SPR may help if the conformation of the σ1R in the absence of interaction attracts ligands such as β-endorphin. Usually, receptors are shifted from resting state to function by interactions with certain molecules (GPCR and G proteins; Gα subunits and GDP/GTP). In the functional conformation they display high affinity for regulatory ligands but not in the resting state. Because ligands exert biased activity to regulate a series of σ1R interactions with other proteins it is probable that in the absence of such interactions physiological ligands display low affinity to this receptor (Front Pharmacol, 10: 634, 2019, PMID: 31249525). 

We agree that further studies are necessary to characterize intimate aspects of β-endorphin 1-31 binding to σ1Rs.

Additionally, some of the figures could be improved/clarified. Figure1

1A. Are the brains shown here matched in treatment with the bar graphs below? If so, the figure may be reconfigured to convey this more clearly, or the legend clarified.

Fig 1A. You are right, each line of brains matches with the bar below. The Figure has been revised accordingly.

1B. The organization of these plots might be changed, so that the data from the wt be laid out in a row of 5 panels, and then the sigma1 KO data for saline and N-acetyl beta endorphin be placed under the corresponding wt data. Also, there are no statistical comparisons performed between wt/N-acetyl beta endorphin and KO/N-acetyl beta endorphin and if appropriate to do so from an experimental perspective, the comparison probably should be made and reported.

The figure 1B has been rearranged following your indications.

Figure2

2A. The data in 2A/row2/left panel shows the * label with a bar bracketing the two doses, which suggests difference between the doses, while the figure legend suggests the * reflects differences with saline. To match the figure legend, both doses of data should be given * labels where appropriate. If differences are observed between the doses, that should be given a different label. Is the ANOVA done for each time point separately? Or as a two-way ANOVA? Also, the comparison could be done as ANOVA of AUC (area under the curve) of the (PWT x time).

The symbols (*) are properly indicate time interval and αN-acetyl β-endorphin 1-31 dose. We also indicate differences between αN-acetyl β-endorphin 1-31 doses (Ï•). Although we handle statistics at the level of essentials, we consult with the Department of Statistics, Center for Informatics at the Spanish Council for Scientific Research (CSIC) on the choice of tests appropriate to the questions asked in the trials. We hope that their choice will provide the statistical confidence necessary for our purpose.

We are using One Way ANOVA covering each assay, doses and evaluation intervals, followed by the All Pairwise Multiple Comparison Procedures (Holm-Sidak method): Overall significance level = 0,05.

Figure 4

In panel 4D, the PRE084 plot should be plotted similar to the beta-endo 1-31 plot with the 100% line drawn lower

in the figure and the legend placed above that line, so that the implied increase of  Gz binding can be highlighted.

Changed

Figure 5

Fig 5A. The top labels and the legend mention G protein changes, but there is no data in 5A to support G protein association changes in this figure panel.

Corrected.

Figure 6.

6A, 6B, and 6F: scale bars not present.

Done

Bottom of Figure6, the 3D protein structures in this figure are of unclear value to this manuscript. In fact, experimentally derived crystal structure have been reported for human sigma-1 (Schmidt Nature 2016 PMID 27042935) and sigma-2 (Alon Nature 2021 PMID 34880501), so what is the insight to be gained from these Novastar structures? Do the authors believe the reported structures are not physiological?

Frankly, we have serious doubts about the functional relevance of structures obtained by crystallization of highly purified proteins. The DNAstar approach (Novafold) does not provide better information. I’ve collaborated with Novafold people to predict protein interaction surfaces based on crystallized models, and the idea failed since their predictions did not coincide with those obtained by residue-directed mutagenesis.

Protein crystal structures correspond to those observed in a low-energy conformation and in absence of interactors. To circumvent this potential inconvenient, actual crystallization studies are performed in the presence of ligands, whether possible. Although, we cannot discard the possibility that the reported crystal structures are indeed formed in the higher-energy physiological environment, to date, no study has demonstrated this situation when σ1R or the σR2 interacts with third partner proteins. As we mentioned before, as a result of such interactions σR1 ligands exert interactor-dependent biased activity (Front Pharmacol, 10: 634, 2019, PMID: 31249525), and this finding suggests that σR1s adopt different conformations when binding to disparate proteins.

Thus, we have removed the DNAstar predicted structures.

Figure 7, 8.

Error bars are not shown. Are these averages of triplicates? If no, claims about increases and decreases cannot be made, especially when the separation from the 100% line is small. The inset figure of the DNASTAR structure should be a panel 7D. To make the claim that the specific residues of beta-endorphin and sigma1 or sigma2 interact (as has been asserted in the text in the figure and also the manuscript text), mutagenesis studies should be performed to confirm their involvement in the behavioral phenotypes and biochemical interactions.

We are including SD of the means.

Our study and previous studies from other laboratories suggest that β-endorphin 1-31 interacts with the epsilon receptor through the region we indicated. In addition, we have conducted studies of SAR with a number of chimeras derived from β-endorphin 1-31 and N-acetyl derivatives and the data support the implication of this region. However, this is not the site to display these data, and this sentence will be eliminated from the panel 7D. We therefore agree that on the basis of the information we provide it is necessary to deepen the more intimate aspects of this union.

Methods:

In section 4.2, infarct area calculation after MCAO, were all 6 slices shown in Fig1A used for the denominator for the area determination? Or was the slice with the greatest infarct volume used for the data shown in 1A? Please clarify.

Infarct area calculation after pMCAO includes the total infarcted brain area as indicated in line 899  ”each side of the coronal sections ….” To improve readability last lines of section 4.2 have been rewritten.

Overall, the data in this manuscript covers an impressive range of behavioral studies and biochemical studies that support the importance of the N-acetyl beta endorphin peptide in physiologic processes, and that the beta endorphin peptide and N-acetyl beta endorphin peptide can modulate interactions of sigma factors with each other and with G proteins and the mu opioid receptor protein. However, some further controls as suggested in the comments above would bolster some of the claims which are made in the manuscript. It is the hope of this reviewer that these controls may be provided for review, and some questions resolved, and perhaps some wording qualified, prior to publication.
